# The bright side of pessimism: Promoting wealth redistribution under (felt) economic hardship

**Silvia Galdi**[1]*, **Anne Maass**[2], **Annalisa Robbiani**[2]

**1** Department of Psychology, University of Campania Luigi Vanvitelli, Caserta, Italy, **2** DPSS, University of Padova, Padova, Italy

* silvia.galdi@unicampania.it

**Data Availability Statement:** All relevant data are within the manuscript and its Supporting Information files.

**Funding:** AM Grant PRIN 2017 #2017924L2B entitled "The psychology of economic inequality".

## Abstract

Economic inequality is a collective issue that affects all citizens. However, people often fail to support redistribution strategies aimed at redressing inequality. In this work we investigated personal optimism and collective pessimism as psychological processes that contribute to hampering vs. promoting the demand for redistribution. Our prediction was that support for redistribution would require both a pessimistic economic outlook at the collective level and the perception of being economically disadvantaged. In two studies, one of which pre-registered, Italian participants (Study 1: $N = 306$; Study 2: $N = 384$) were led to feel relatively poor or rich, rated their perceived control over either their personal or the nation's future and estimated either personal or national economic and general future risks. To measure support for redistribution, participants were invited to allocate their desired level of taxation to each of the five tax brackets included in the Italian personal income tax. Results showed that participants were optimistic about their personal future, but pessimistic about the fate of their nation. This difference was explained by respondents' greater perceived control over personal future than over the nation's future. Importantly, greater pessimism about national economic risks led to greater support for progressive taxation only for participants who felt relatively poor.

## Introduction

At a time when in practically all developed countries wealth and income are increasingly and disproportionately concentrated in the hands of fewer and fewer millionaires [1–4], one would expect that the majority of people should not hesitate to endorse social programs aimed at redressing the economic gap between the most and the least privileged of society [5]. However, this is not the case [6–8]. Despite widespread agreement that wealth and income should be distributed more fairly [9–11], people often fail to support concrete redistribution strategies, such as more progressive taxation [12, 13].

Among the different strategies proposed by economists and organizations alike to reduce the economic gap, progressive taxation occupies a central place. For instance, the 2020 annual report of Oxfam, entitled "Public Good or Private Wealth", proposes to "end the under-taxation of rich individuals and corporations", to "tax wealth and capital at fairer levels", to "stop

Italian Ministry of Education, University and Research (MIUR). The funder had no role in study design, data collection and analysis, decision to publish, or preparation of the manuscript.

**Competing interests:** The authors have declared that no competing interests exist.

the race to the bottom on personal income and corporate taxes", and to "eliminate tax avoidance and evasion by corporations and the super-rich" [14]. Likewise, the World Economic Forum recently argued that "the introduction of a tax on passive income and wealth is essential in a world where individuals are wealthier than nations" [15]. And the International Monetary Fund has recently argued that the Covid-19 pandemic has made it ever more important "to move towards a fairer and more equitable taxation of economic activities at the global level" [16]. Although progressive taxation is considered a pillar of redistribution by many experts, lay people, though desiring a fairer society at an abstract level, rarely support increasing progressivity of taxation for income, corporate, or inheritance. Thus, there is a remarkable contrast between the abstract quest for reducing inequality and the failure to support concrete redistribution strategies such as progressive taxation [17], which is the focus of the present research. Ironically, those social classes that would benefit most from such concrete redistribution strategies are often particularly reluctant to support them [18].

Therefore, a challenging question for researchers, is to understand why people, especially those at the low end of the social and economic ladder, may be so reluctant to stand up for concrete redistribution strategies, even though they would potentially benefit most from such changes. What are the psychological processes that hamper (vs. promote) the demand for redistribution and what communication strategies could be employed to overcome such barriers, especially among those at the low end of the social and economic ladder?

Psychological literature offers a number of explanations. First of all, people may be unaware of actual wealth or income gaps, greatly underestimating their country's level of inequality [19, 20; but see 21, for different results]. This occurs, in part, because the immediate environment may bias people's estimates. To form their judgements individuals typically observe wealth or income levels of their local community [22, 23], or of their idiosyncratic social networks [24], and then infer the entire distribution from that limited information. However, providing accurate information about existing wealth or income gaps is not always sufficient to overcome resistance to redistribution policies [25].

Thus, additional psychological factors are likely to be at play. Among these figure the belief in a just and meritocratic world [26, 27], the attribution of poverty and wealth to individual rather than structural characteristics [28], the desire to defend and maintain existing social hierarchies [29], and the motivation to justify the system that produces these inequalities, but on which people's livelihoods depend [30, 31]. While all these factors matter, another variable may contribute to reducing willingness to support redistribution policies: people may be overly optimistic about their (and their children's) prospect of climbing the social and economic ladder in the future, which, in turn, makes wealth and income gaps appear more tolerable [32].

Especially in the U.S., for example, citizens stubbornly overestimate the likelihood that a person might rise-up, but not move down, the social and economic ladder [33–38; but see 39, 40]. Although current levels of social mobility are particularly low in the U.S. compared to other developed countries [41], this evidence suggests that Americans have a strong faith in the "American Dream" promise, according to which all people have equal opportunities and can improve their social and economic status if they are determined to work hard. These optimistic beliefs about socio-economic advancement, however, appear to be a uniquely American phenomenon, given that Europeans, for instance, tend to be more realistic or may even underestimate social mobility in their nations [26, 41]. Nevertheless, studies have documented that views on social mobility affect policy preferences across all countries, with greater optimism being associated with lower support for redistribution [42–44].

The aim of the present work is to take this argument one step further by showing for the first time that optimism in general, and not only concerning social mobility, contributes to hampering the demand for redistribution. The starting point of our argument is that with

increasing economic inequality people tend to focus more on the individual [45]. This focus on the self rather than on the collective may make people unrealistically optimistic, thus discouraging them from demanding redistribution. On the contrary, we will demonstrate experimentally that a pessimistic economic outlook at the collective level, combined with the feeling of being disadvantaged, can motivate individuals to embrace redistribution policies.

A pessimistic economic outlook is typically interpreted as undesirable by economists, as it encourages precautionary saving at the expense of spending [46–48]. Overall, indeed, when their views about the personal and national economic situation (commonly referred to as consumer confidence) are optimistic, people tend to increase spending; conversely, when these same views become pessimistic, expenditure, which many experts consider the driving force of any national economy, decreases [46].

Unlike economists, social psychologists, at least under some circumstances, can look positively at a pessimistic outlook, in that it may be the driving force of social change. For instance, according to relative deprivation theory [49], to predict responses to wealth and income inequalities we must consider people's subjective interpretation of their lot in life. The theory posits that feelings of relative deprivation depend on three types of social comparisons that people make: interpersonal (whereby people feel that they are personally deprived relative to others), intergroup (whereby people feel that their ingroup is deprived relative to another group), and past/present comparisons (whereby people feel that they, or their ingroup, are deprived relative to the past). Research has long demonstrated that feelings of relative deprivation, especially those based on intergroup comparisons, increase the likelihood of collective action aimed at promoting social change and redressing inequality [50–52]. The relation between relative deprivation based on intergroup comparisons and collective action targeting redistribution policies, however, is by no means linear and can be moderated by third variables, such as system-justifying beliefs [53, 54]. Nonetheless, requests for redistribution are difficult to imagine unless one feels deprived and is pessimistic about the future.

In line with this contention, findings concerning environmental activism have shown that optimistic messages about carbon emissions reduce the motivation to engage in mitigation, whereas pessimistic messages are motivating, as they portray a negative reality in need of change [55]. The same principle seems to apply to support for redistribution policies in the economic realm. For instance, Ravallion and Lokshin analyzed responses of 7000 Russian adults surveyed in 1996 [56]. According to these survey data, support for governmental redistribution was stronger among citizens who put themselves on the low end of the social and economic ladder and expected their welfare to decline in the near future. Support for redistributive social policies was found to be higher even among those respondents who were wealthy according to objective indicators but perceived that their economic situation had deteriorated relative to the past and, at the same time, were pessimistic about the future. By contrast, a negligible demand for redistribution emerged among citizens who perceived themselves to be well-off and expected to be better-off in the future.

Similar findings have been recently obtained by Garcia-Muniesa [57], who analyzed data from a large-scale survey conducted in nine European countries (including Italy). Results demonstrate that having personally suffered worsened economic conditions during the global financial crisis of 2008 was associated with greater support for progressive taxation only for citizens whose economic prospects for the future had become pessimistic. Conversely, citizens who considered that the economic shock they had experienced in 2008 was temporary and were optimistic about the economic situation in the near future did not show support for progressive taxation. Overall, these findings show that only people who feel deprived and believe that things will get worse support redistribution, probably because they expect to benefit the most from redistributive policies.

On the basis of this correlational evidence, therefore, two preconditions appear to be needed for triggering individual support for redistribution, including progressive taxation. Drawing from the concept of relative deprivation, the first precondition is dissatisfaction with current personal economic situation, regardless of one's objective socio-economic status. The second crucial precondition is a person's expectation for the future: only a pessimistic economic outlook seems to motivate people to demand social programs aimed at redressing wealth and income inequalities. At the same time, however, these results raise an interesting question. How is it possible that many people continue to maintain a positive outlook on themselves and their future prospects even in the light of increasing wealth and income inequalities, of the relative impoverishment of a large part of the population, and after experiencing major economic crises such as the one of 2008?

## Personal optimism, collective pessimism

One possible answer to this intriguing question comes from recent research showing that wealth and income inequalities not only have important implications for the well-being of society [58–61] and undermine a nation's economic output (which further deepen these inequalities [62, 63]), but also trigger a competitive and individualistic normative climate. For example, across three experiments with samples from different countries (i.e., Spain, Australia and the United States), Sánchez-Rodríguez and collaborators manipulated the degree of economic inequality in a fictional society [45]. Results show that high economic inequality enhanced the perception that people in society are more independent, more involved in exchange relationships, more likely to prioritize their personal goals, more competitive and less cooperative. In contrast, low economic inequality triggered the perception that societal norms encompass interdependence, involvement in communal relationships, and the prioritization of group goals. Therefore, with increasing of economic inequality, people tend to focus more on themselves rather than on the collective. This matter is important. The focus on the self may indeed make people unrealistically optimistic, possibly discouraging them from demanding redistribution.

Starting from Weinstein's ground-breaking study [64], it is well established that when thinking about their own personal future (e.g., personal vision of their lives), as well as about the future of their own families, most people are not objective in their prediction. Rather, they tend to be deeply and resolutely optimistic, believing that their own chances of experiencing a negative event are lower (or a positive event higher) than can possibly be true [65]. For example, people underestimate their own risk of getting involved in an automobile accident [66], or the risk of being personally afflicted by various diseases or health problems [67–69], but overrate the longevity of their own dating relationships [70], or their children's chance of staying happy [71].

This cognitive illusion is known as unrealistic optimism and is defined as the difference between a person's expectations and the outcome that follows. If expectations are better than reality, optimism is unrealistic. To date, a number of studies have consistently reported that most people have personal expectations that are better than reality [72], and that individuals continue being optimistic even in the face of disconfirming evidence [73]. Unrealistic optimism, moreover, is a global phenomenon: it has been observed in many different countries, in Western and in non-Western cultures, in women and men, in children and in the elderly [65, 74].

Although people are optimistic about themselves, their children, and their family, the majority of individuals tend to feel pessimistic about the fate of their fellow citizens, the fate of their nation, and about the possibility to address important collective problems, such as

environmental pollution or wealth and income inequalities [75]. For instance, Wenglert and Rosen [76] found that people were much more optimistic about their personal future than about the world's future [see also 77]. The authors also showed that optimism about personal future was only weakly associated to the perception of the world's future, thus suggesting that unrealistic optimism and pessimism about the collective are independent. For simplicity, in the present work we will refer to the two phenomena as *personal optimism* and *collective pessimism*, respectively.

The asymmetry between personal optimism and collective pessimism has been confirmed also by large-scale surveys concerning people's perceptions of the economic situation. In the 2011 Eurobarometer Survey [78] respondents were asked about their personal and general economic outlooks. On the personal level, around 56% of Europeans expected that their financial situation would remain the same in the next future, while 20% expected their situation to improve. Very similar results emerged for respondents' expectations with respect to their personal job situation. At the same time, however, most Europeans (41–44%) thought that the economic situation in their home country, as well as in the European Union and in the world, would get worse.

In a similar vein, other surveys indicate that people in developed countries tend to be rather pessimistic about the future of their nations, with Italians showing a particularly high level of pessimism [79]. For instance, in Italy, where wealth and income inequalities have increased disproportionately over the last ten years [80], 54% of citizens continue to show personal optimism, believing that their living conditions will not change or even improve. At the same time, however, 64% of the population expects that the nation's economy will get worse [81]. Similarly, most Americans predict a rather bleak future for the nation, expecting a weaker economy, an increasing income gap, and a broken political system [82]. Yet, they continue to be very optimistic about their personal financial situation, with over 70% expecting their finances to improve [83].

Overall, these findings allow us to draw two important conclusions. First, people reason quite distinctly about themselves and the collective, showing a very stable optimism regarding the future of their personal situation but pessimism about the future of the collective (i.e., their country or the world). Second, especially in countries with high levels of economic inequality people tend to focus more on themselves rather than on the collective. This confirms that, making them unrealistically optimistic, the focus on the self may trigger people's fear of being hurt by redistribution policies in the future, thus hampering the demand for such policies. Conversely, making them pessimistic by shifting their focus on the collective may be a crucial (but still under-researched) requirement for people to support redistribution.

## The role of perceived control

The reasons for the asymmetry between personal optimism and collective pessimism may be best understood by analyzing the likely underlying psychological mechanisms. There are many cognitive and motivational reasons for why people maintain an exaggerated positive outlook at their personal future even in the face of disconfirming evidence. These include selective information processing (beliefs are updated faster in response to positive than to negative information [65]), differential mentalizing (positive future events are envisaged in greater detail and as closer in time than negative events [65]), and the fact that being optimistic about the future may be adaptive, as it reduces stress and anxiety, ultimately benefitting physical and psychological health and survival [84]. All these mechanisms contribute to producing personal optimism that is resistant to change. An additional process is a person's perceived control over outcomes [85], which, we believe, is particularly informative not only about personal optimism per se, but also about the asymmetry between personal optimism and collective pessimism.

It has long been argued that personal optimism [64] is driven mainly by the (illusory) perceived personal control over events [86, 87]. In line with this interpretation, personal optimism is particularly strong for events that are believed to be under personal control. For instance, people tend to show greater personal optimism about events they perceive to be more controllable, such as car accidents or alcohol and drug dependence, but less so about events such as heart attack, cancer, or being victimized by burglary, that are typically perceived as uncontrollable, or more difficult to prevent [64, 88]. Moreover, research has shown that personal optimism is particularly pronounced in countries such as the U.S. and Canada, where personal responsibility and control are deeply ingrained in culture [87, 89]. Thus, both comparisons between types of risks (i.e., controllable vs. uncontrollable) and between countries confirm that personal optimism is, in large part, a reflection of underlying (illusory) control beliefs.

By extension, perceived personal control offers also a plausible explanation for the difference between personal optimism and collective pessimism. It is logical that people envisage to have greater control over their personal fate than over that of their nation or the world (i.e., the collective). Hence, if the driving force of personal optimism is perceived personal control over events, then collective pessimism may be due to the fact that people view events of a collective character as only to a minimal extent under their personal control. In line with this assumption, it has been demonstrated that collective pessimism increases with increasing spatial distance between the object of evaluation and the self. For instance, people are more optimistic about the future of their local community than about that of their nation or the world [90]. If we assume that personal control decreases with increasing spatial distance from the self, these findings would support the explanation based on perceived personal control. Together, the above literature suggests that the difference between personal optimism and collective pessimism may be explained, at least in part, by the fact that people perceive greater personal control in the former than in the latter case.

Building on this evidence, in two studies we investigated for the first time two interrelated issues, namely the role of personal control in personal optimism and collective pessimism, and their relation to demand for redistribution among people who feel relatively economically disadvantaged vs. well-off.

## Overview of research

Redistribution of wealth can be achieved through many different means, such as caps on incomes, limits on transfer of wealth across generations, and the like. One way to reduce wealth and income inequalities is to raise more tax revenue from those most able to pay. However, despite rising levels of wealth and income inequalities, in many developed countries the richest are still undertaxed [91] and public support for redistribution through more progressive taxation is very low [92]. For example, in Italy the top rate of personal income tax is 43% compared to 72% in 1975, and current legislative proposals aim at further reducing or even eliminating progressivity of personal income tax (i.e., flat tax). Therefore, to investigate support for concrete redistribution policies aimed at redressing wealth and income inequalities, in the present work we focused on progressive taxation. Specifically, we invited participants to allocate their desired level of taxation to each of the five tax brackets included in the Italian personal income tax (i.e., the IRPEF).

Our main argument is that specific conditions must be met for more progressive taxation to become a desirable goal. Specifically, support for more progressive taxation requires both the perception of being economically disadvantaged and a pessimistic economic outlook at the collective level. In order to investigate these two mechanisms and their interactive effects, we pursued two goals. First, we aimed at showing that the asymmetry between personal optimism

and collective pessimism is explained in large parts by perceived personal control. To test the role of personal control, we activated either a personal or collective mind-set and invited participants to rate their personal control over either their own future (personal mind-set) or the nation's future (collective mind-set). We then assessed personal optimism and collective pessimism by asking participants to estimate either personal or collective expected economic risks (e.g., risks related to future economic problems or future salaries). In addition to specific economic risks we also assessed general risks both at the personal (e.g., the risk of getting involved in a car accident) and collective level (e.g., the risk that a powerful earthquake will hit Italy). We expected to find personal optimism and collective pessimism both for economic and general risks. However, we predicted that only economic (but not general) risks would be related to redistribution. Our specific hypotheses were that:

**Hypothesis 1.** Participants would perceive greater control over their own future than over the nation's future.

**Hypothesis 2.** Participants would judge expected economic (Hypothesis 2a) and general risks (Hypothesis 2b) less likely for themselves, but more likely for Italy.

**Hypothesis 3.** Differences in perceived control would account for the different levels of optimism at the personal vs. collective level (mediational hypothesis).

The second aim of our work was to investigate *whether* and *when* collective pessimism translates into the demand for more progressive taxation. We assumed that a pessimistic economic outlook at the collective level (i.e., collective economic pessimism) would trigger a preference for more progressive taxation only among individuals who feel economically disadvantaged in the society. As discussed in the introduction section, considerable evidence shows that redistributive policies are supported more strongly not only by citizens who actually are, but also by those who feel economically disadvantaged [93–95]. Since poor and rich people differ on an unknown number of variables other than wealth and income, in the present work we decided to manipulate relative economic hardship vs. prosperity experimentally. We restricted our samples to individuals with the same income range (1200–1800 Euros per month) and invited participants to provide information about their socio-economic situation. Then, we led participants to feel relatively poor or well-off, by informing them that, on the basis of the collected information, their socio-economic status was low vs. high. We therefore predicted that:

**Hypothesis 4.** Support for progressive taxation would be an interaction function of feeling relatively poor (vs. well-off) and being in a collective (vs. personal) mind-set. Thus, greater pessimism about collective (vs. personal) economic risks would lead to greater support for progressive taxation only for participants who felt relatively poor.

**Hypothesis 5.** Collective mind-set would reduce participants' perceived personal control, which, in turn, should increase pessimism about collective economic risks. This greater pessimism about collective economic risks would lead to greater support for progressive taxation, but only for participants who felt relatively poor. Our working model is represented in Fig 1.

## Study 1

The experiment was created using the software Survey Monkey and designed in such a way as to avoid any missing data. The experiment consisted of 2 X 2 factorial design. The independent variables of interest were the manipulated socio-economic status (low or high) and the mind-

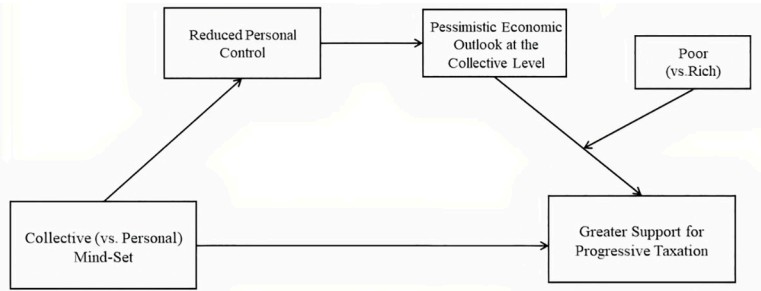

**Fig 1. Key paths of the working model.**

set (personal or collective), which were manipulated between-participants to obtain 4 levels (low socio-economic status and personal mind-set, high socio-economic status and personal mind-set, low socio-economic status and collective mind-set, high socio-economic status and collective mind-set) resulting in 4 surveys. The main dependent variables consisted of the two brief scales aimed at assessing participants' personal control and expected risks, and the taxation task.

## Method

### Participants

Residents of Southern Italy were contacted by one of eight (4 women and 4 men) experimenters involved in data collection through snowball sampling (including neighbors, acquaintances, etc. [see 96, for a similar procedure]) and asked to take part in a research aimed at investigating individuals' beliefs about future everyday life events. Those who agreed to participate in the study were scheduled for an appointment and then visited by the experimenter at their homes, where they completed the relevant tasks individually on a laptop computer. A total of 355 participants completed the survey. The sample consisted of 159 men (45%) and 192 women (55%), including 305 (87%) employees, 43 (12%) self-employed, and 3 (1%) retired persons. Participants were all Italians and their age ranged from 27 to 72 years ($M_{age}$ = 45.79, $SD$ = 10.57). Most respondents had a high school diploma ($n$ = 153; 44%) or a university degree ($n$ = 160; 45%), whereas 38 participants (11%) had the lowest formal qualification. All participants had a personal monthly income ranging from 1200 to 1800 euros. We decided to limit the personal monthly income of our sample to this range because, according to a recent national study [97], Italians' average monthly income is around 1500 euros. Importantly, we believed that the more homogeneous our sample was with respect to monthly income range, the more the manipulation of participants' socio-economic status would be likely to be effective, as this allowed us to avoid the potentially confounding variables that typically covary with individuals' actual socio-economic status (e.g., political orientation, tax attitudes, system justification). Recruits volunteered in the experiment without monetary compensation. Participants performed the relevant tasks in the same order as presented below.

 **Procedure.** After consenting to participate in the study, respondents filled out one of the 4 surveys, depending on the randomly assigned condition. The surveys were divided into three sections and included the same measures in the same order. In Section 1, participants rated their current mood and provided information about gender, age, level of education, and profession. To assure a fairly homogeneous sample with respect to income, our plan was that only respondents with a personal monthly income between 1200 and 1800 euros would be eligible

for the study. Therefore, participants were invited to indicate their personal monthly income on one of two 5-income range scales. Both scales included the "1200–1800 euros" range. However, depending on the manipulated socio-economic status condition to which participants had been assigned to, the "1200–1800 euros" range was the second (i.e., low socio-economic status condition) or the fourth (i.e., high socio-economic status condition) income range of the scale. If a participant marked the "1200–1800 euros" income range, continued the experiment. Conversely, if a participant selected a different income range, the survey would be interrupted, the participant fully debriefed, thanked and then dismissed. Subjects eligible for the study then provided data on family size, composition and basic characteristics of its current members. Next, they were informed that the software would calculate their socio-economic status on the basis of the collected information. Immediately afterwards, participants were shown a picture of a social ladder with 10 rungs and learned that the software had placed them on the third (i.e., low socio-economic status condition) or the seventh rung (i.e., high socio-economic status condition) of the social ladder.

In Section 2 of the survey, we introduced the second independent variable. Given that, presumably, individuals' personal mind-set is highly salient when individuals are asked to think of their own lives, whereas a collective mind-set is highly salient when asked to think of their nation, we invited respondents to fill out three items aimed at assessing beliefs of personal control over either one's own future (personal mind-set condition) or one's nation's future (collective mind-set condition). After completing a question about perceived influence of nation on self, participants were presented with a list of seven possible events that could either happen to a person or to a nation, depending on the mind-set condition to which they had been assigned to. Subjects rated their own chances (personal mind-set condition) or their nation's chances (collective mind-set condition) of experiencing each event in the next twenty years compared to the average peer (vs. the average OCSE nations).

In Section 3, participants were instructed to allocate the percentage of taxation they regarded as the ideal to each of the five tax brackets included in the Italian income tax called IRPEF (taxation task). Finally, respondents were asked to remember which place on the social ladder they had been assigned to by the software at the beginning of the experiment, rated their personal agreement with assigned socio-economic status, and evaluated again their current mood. At the end of the experiment, participants were fully debriefed and thanked for their participation. Prior to dismissal, all subjects were given the opportunity to either withdraw their data or sign a release form. All participants signed the form. Each experimental session took about 15 minutes. The procedure and materials of the study had been approved by the Ethics Committee for Psychological Research of the University.

## Materials

**Mood.** Mood was assessed at two measurement times: at the very beginning (Time 1) and at the end (Time 2) of the survey. Participants indicated how they felt at that moment using a scale ranging from 1 (*bad*) to 10 (*very good*).

**Manipulated socio-economic status.** Participants were asked to indicate their personal monthly income on one of two scales, which were used for a twofold objective: a) excluding from the sample participants with a personal monthly income below 1200 and above 1800 euros and b) introducing the manipulation of participants' socio-economic status that would be realized subsequently. Each scale included 5 income ranges. However, in the low socio-economic status condition, the 1200–1800 euro-range was the second (i.e., up to 1200 euros, 1200–1800 euros, 1800–2400 euros, 2400–3000 euros, more than 3000 euros), whereas in the high socio-economic status condition it was the fourth euro-range of the scale (i.e., 0–300

euros, 300–700 euros, 700–1200 euros, 1200–1800 euros, more than 1800 euros). Moreover, to increase participants' focus on the place of their personal income on the scale, a different background color was used to highlight each of the 5 ranges, with nuances varying from red (first and lower income range), over orange and yellow (second and third), to light and dark green at the upper part of the scale (fourth and fifth income range). Therefore, in the low socio-economic status condition the background of the 1200–1800 euro-range was colored orange, whereas in the high socio-economic status condition it was colored light green. The manipulation is reported in S1 Fig. After indicating their personal monthly income, participants provided information on family size, composition of its current members (i.e., number of family members below 18 years of age) and the number of people contributing to the family income. Immediately afterwards, respondents read the following: "On the basis of information you have just provided, the software is calculating your socio-economic status". Participants were then shown a picture portraying a social ladder with 10 rungs that was described as follows: "Think of the ladder below as representing where people stand in your country. At the top of the ladder (10th rung) are the people who are the BEST off—those who have the MOST money, the MOST education and the MOST respected jobs. At the bottom (1st rung) are the people who are the WORST off—those who have the LEAST money, the LEAST education and the LEAST respected jobs or are unemployed". Then, participants were presented with the same social ladder including a big red dot positioned either on the third (low socio-economic status condition) or the seventh rung (high socio-economic status condition) and were told: "The software has calculated your socio-economic status. As you can see in this figure, you have been placed on the third (vs. seventh) rung of the social ladder. The red dot shows your position".

**Personal control.** Right after the manipulation of participants' socio-economic status, we manipulated participants' mind-set. Depending on the mind-set condition to which participants had been assigned to (i.e., personal vs. collective), three items were used to trigger a focus on the self or the collective, and to assess beliefs of personal control. In the two experimental conditions, the items were identical with the exception that a) in the personal mind-set condition they were aimed at leading participants to focus specifically on themselves and their own personal future (Item 1a: "I am convinced that what will happen to me in the future depends on me"; Item 2a: "I can do a lot to change important aspects of my life"; Item 3a: "What happens in my life does not depend on me"), whereas b) in the collective mind-set condition the three items were aimed at leading participants to focus specifically on their nation's future (Item 1b: "I am convinced that what will happen to Italy in the future depends also on me"; Item 2b: "I can do a lot to change important aspects of Italy"; Item 3b: "What happens in Italy does not depend on me"). Responses were provided on scales ranging from 1 (*strongly disagree*) to 7 (*strongly agree*). Single averaged scores of personal control were calculated after reverse-coding Item 3a and Item 3b (Cronbach's $\alpha$ = .69). For all participants, therefore, higher values reflect greater beliefs of personal control.

**Perceived influence of nation on self.** The perceived influence of the nation on the individual was assessed using a single item (i.e., "What happens in Italy does not change my life"). Participants rated how much they agreed with the statement on a scale ranging from 1 (*strongly disagree*) to 7 (*strongly agree*). Responses were then recoded so that high values of perceived influence indicated that participants feel personally more affected by what happens in their country.

**Expected risks.** When studying personal optimism, the more common strategy is to measure comparative rather than absolute risk estimates [72]. The aim of the comparative strategy is to find out whether people think that their risk of experiencing a negative event is lower (or higher) than the average peer's risk, not whether their risk is lower (or higher) than the actual

risk. To investigate comparative risk estimates, participants may be asked for a "direct" comparison, in which they judge whether their own risk is smaller, greater, or the same as their peers' risk, or for an "indirect" comparison, providing separate risk estimates for themselves and others. In the present work, we assessed personal optimism and collective pessimism using the "direct" comparative method. Therefore, we created a brief scale including 7 hypothetical events referring, depending on condition, either to personal (the individual participant) or to collective risks (the nation). Of these, 4 events, which were of primary interest to our study, were *economic risks*. These events were identical, but framed so as to refer either to personal or collective risks (i.e., personal economic risks: "Being able to manage economic problems"; "Needing to ask for a loan to pay off a debt"; "Getting a very good salary"; "Job loss or getting fired"; collective economic risks: "Number of families in Italy with economic problems"; "Number of families in Italy asking for a loan to pay off a debt"; "Number of families in Italy relying on a very good salary"; "Number of families in Italy without income because of job loss or dismissal"), depending on the randomly assigned mind-set condition. The remaining 3 events were, depending on condition, either personal (i.e., "Getting divorced"; "Being in a car accident"; "Developing drinking problems") or collective (i.e., "A grave flood hitting Italy"; "A powerful earthquake hitting Italy"; "Solving problems of waste in Italy") *general* risks. The two versions of the scale are reported in S1 File. For each event, respondents indicated the probability that it may happen to them personally (vs. Italy) in the next twenty years, compared to the average of persons of their same age and sex (vs. the average of OCSE nations) on scales ranging from -3 (*absolutely below average*) to +3 (*absolutely above average*), with 0 as the midpoint (*within the average*). Thus, participants either rated each risk for themselves compared to other Italians or for Italy compared to other OECD countries. One item related to general risks (i.e., "Solving problems of waste in Italy") and 3 items related to economic risks (i.e., "Successfully managing economic problems"; "Getting a very good salary"; "Number of families in Italy relying on a very good salary") referred to positive events and were reverse coded. Scores were then averaged into single indexes of economic risks (Cronbach's α = .73) and general risks (Cronbach's α = .64). Therefore, lower negative scores of economic risks and general risks reflect greater optimism about the future (chances of negative outcomes below average).

**Taxation task.** To investigate the impact of manipulated socio-economic status (low, high) and mind-set (personal, collective) on support for actions to reduce disparities in the distribution of income and wealth, we asked participants to perform a taxation task. Before the task, we ensured that all participants had the same working definition of the Italian personal income tax (IRPEF). Participants were exposed to a table showing the current five tax brackets of the IRPEF, the tax rate for each of the 5 brackets, and a brief practical example aimed at helping participants to understand how IRPEF works. The table was described as follows: "According to the current Italian tax system, Italians are subjected to a progressive personal income tax called IRPEF, which includes five tax brackets. As you can see in the example, each of the five tax brackets is subjected to a tax rate that increases with increasing of one's personal income, and ranges from 23% (for the first tax bracket) to 43% (for the fifth tax bracket) of the personal income". Immediately afterwards, participants were presented with five rows representing the current five tax brackets of the IRPEF, and required to allocate, for each bracket, the tax rate that they regarded as ideal. Single scores of progressivity of taxation were then calculated by subtracting the tax rate assigned to the first and lowest bracket from the tax rate allocated to the fifth and higher tax bracket. Therefore, higher values of the index reflect greater progressivity of self-generated taxation.

**Manipulation check and personal agreement.** Participants were asked to remember which place on the social ladder they had been assigned to by the software at the beginning of the experiment (manipulation check), using a scale ranging from 1 (*first rung*) to 10 (*tenth*

*rung*). Respondents also rated their personal agreement with assigned socio-economic status (personal agreement) on a scale ranging from 1 (*not at all*) to 5 (*entirely*).

## Results

### Preliminary analyses

In the taxation task, 4 participants consistently indicated a 0 tax rate for all 5 tax brackets. Given that these responses were clearly anomalous, these participants were excluded from the analyses. Additional 45 participants failed one of the two manipulation checks: 4 participants did not correctly remember which place on the social ladder they had been assigned to (manipulation check), whereas 41 participants (12%) did not agree at all with the assigned position (personal agreement), thus suggesting that the manipulation of socio-economic status was not effective. After excluding these 49 participants, an ANOVA was conducted on personal agreement, using manipulated socio-economic status (low, high) as independent variable. On average, participants showed a fair amount of agreement with the assigned socio-economic status (*M* = 2.88, *SD* = .81). Average agreement with the assigned condition was not reliably different in the low socio-economic status (*M* = 2.83, *SD* = .81) and the high socio-economic status condition (*M* = 2.92, *SD* = .81; *F* < 1.05, *p* > .30), thus suggesting that the manipulation of participants' socio-economic status had worked as intended. Therefore, analyses were conducted on a final sample of 306 respondents.

Participants were approximately equally distributed across the four conditions (low socio-economic status and personal mind-set, high socio-economic status and personal mind-set, low socio-economic status and collective mind-set, high socio-economic status and collective mind-set) in terms of gender, age, education, and occupation (all *ps* >.15). Zero-order correlations among participants' family characteristics (family size, number of family members below 18 years of age, number of people contributing to the family income) and main study variables (mood Time 1, mood Time 2, perceived influence, personal control, economic risks, general risks, progressivity of taxation) are presented in Table 1. Participants' family size correlated positively with the number of family members below 18 years of age and the number of people contributing to the family income. However, no relation emerged among these variables and

**Table 1. Study 1 (*N* = 306).** Zero-order correlations among study variables (Family Size, Number of Minors in the Family, Number of People Contributing to the Family Income, Mood Time 1, Mood Time 2, Perceived Influence, Personal Control, Economic Risks, General Risks, and Progressivity of Taxation).

| Variables | Correlations | | | | | | | | | |
|---|---|---|---|---|---|---|---|---|---|---|
| | 1 | 2 | 3 | 4 | 5 | 6 | 7 | 8 | 9 | 10 |
| 1. Family Size | – | | | | | | | | | |
| 2. N° of Minors | .433*** | – | | | | | | | | |
| 3. N° People Contributing | .388*** | -.110* | – | | | | | | | |
| 4. Mood Time 1 | -.035 | .031 | -.005 | – | | | | | | |
| 5. Mood Time 2 | -.031 | .012 | -.008 | .817*** | – | | | | | |
| 6. Perceived Influence | -.015 | -.028 | -.030 | -.117* | .075 | – | – | | | |
| 7. Personal Control | .053 | .004 | -.027 | .060 | .076 | -.174** | | | | |
| 8. Economic Risks | -.089 | -.035 | -.030 | .102 | -.210*** | -.140* | -.318*** | – | – | |
| 9. General Risks | -.061 | -.093 | .013 | -.142** | -.176** | -.148 | -.163** | .652*** | | |
| 10. Progressivity of Taxation | -.067 | -.003 | -.031 | .115* | .068 | .010 | -.085 | .119* | .085 | – |

***\*\*\****p* < .001
*\*\**p* < .01
*\**p* < .05.

either personal control, or economic risks, general risks, progressivity of taxation, and personal agreement (all $ps > .25$), such that we will not discuss participants' family characteristics further. Interestingly, greater personal control was associated with higher optimism concerning both economic and general risks.

Descriptive statistics for mood Time 1, mood Time 2, perceived influence, personal control, economic risks, general risks, and progressivity of taxation across experimental conditions (manipulated socio-economic status, mind-set) are presented in Table 2.

Before conducting our main analyses, we investigated the effects of participants' sociodemographic characteristics on main study variables. A series of multiple regression analyses were conducted on personal control, economic risks, general risks, and progressivity of taxation. All regression models included the main effects of manipulated socio-economic status (low = 0, high = 1), mind-set (personal = 0, collective = 1), gender, age, education, and occupation in the first block; the second block included the two-way interactions between each independent variable (i.e., manipulated socio-economic status, mind-set) and each sociodemographic variable. The model predicting participants' scores of personal control revealed a significant Education x Mind-Set interaction effect, $b = .25$, $t(290) = 2.72$, $p = .007$: in the collective, but not personal, mind-set participants with the lowest formal qualification perceived less personal control ($M = 2.87$, $SD = 1.55$) than those with a high school diploma ($M = 4.13$, $SD = 1.41$, $t(93) = -3.49$, $p = .001$) and those with a university degree ($M = 4.47$, $SD = 1.43$; $t(182) = -4.29$, $p < .001$), whereas no differences in perceived personal control emerged between participants with a high school diploma and those with a university degree ($t(137) = -1.43$, $p > .15$). However, participants with the lowest formal qualification, as well as those with high school diploma and a university degree reported greater personal control in the personal than in the collective mind-set condition (lowest formal qualification: $t(33) = 3.26$, $p = .003$; high school diploma: $t(135) = 4.76$, $p < .001$; university degree: $t(132) = 2.07$, $p = .04$), thus suggesting that the manipulation was effective regardless participants' level of education. No other interaction effect emerged ($ps > .08$). Therefore, participants' sociodemographic characteristics will be not discussed further.

**Mood.** A 2 (mind-set: personal, collective) x 2 (manipulated socio-economic status: low, high) x 2 (mood: Time 1, Time 2) ANOVA with repeated measures on the last factor was conducted. Participants were in a better mood before (Time 1: $M = 6.53$, $SD = 1.84$) than after (Time 2: $M = 6.32$, $SD = 1.85$) participating in the study, $F(1, 302) = 11.73$, $p = .001$, $\eta^2_p = .04$. Also, they were in a better mood in the personal than in the collective mind-set condition (see Table 2), $F(1,302) = 6.09$, $p = .014$, $\eta^2_p = .02$, as well as in the high ($M = 6.70$, $SD = 1.52$), as compared to the low socio-economic status condition ($M = 5.87$, $SD = 2.11$), $F(1, 302) = 7.39$, $p = .007$, $\eta^2_p = .02$. Most importantly, a significant interaction between mood and socio-economic status emerged, $F(1, 302) = 13.16$, $p < .001$, $\eta^2_p = .04$: being told that one belongs to a relatively low socio-economic status reduced mood from Time 1 ($M = 6.32$, $SD = 2.09$) to Time 2 ($M = 5.87$, $SD = 2.11$), $t(138) = 4.29$, $p < .001$, whereas being told that one belongs to a relatively high socio-economic status had no effect on mood (Time 1: $M = 6.69$, $SD = 1.59$; Time 2: $M = 6.70$, $SD = 1.52$; $t(166) = -.08$, $p = .93$).

**Perceived influence of nation on self.** An ANOVA was conducted on participants' scores of perceived influence, using manipulated socio-economic status (low, high), mind-set (personal, collective), and their interaction as independent variables. Results showed a significant Manipulated Socio-Economic Status x Mind-Set interaction effect, $F(1, 302) = 9.51$, $p = .002$, $\eta_p^2 = .03$. As shown in Table 2, for participants made to feel relatively poor, the perceived influence of the nation on the individual did not reliably differ between personal and collective mind-set, $t(137) < .90$, $p > .39$. In contrast, participants assigned to the high socio-economic status and personal mind-set condition, perceived less influence of the nation on self as

**Table 2. Study 1 ($N$ = 306).** Means and standard deviations of Mood Time 1, Mood Time 2, Perceived Influence, Personal Control, Economic Risks, General Risks, and Progressivity of Taxation as a function of Manipulated Socio-Economic Status (Low SES, High SES) and participants' Mind-Set (Personal, Collective).

| Variables | | Mind-Set | |
|---|---|---|---|
| | | Personal | Collective |
| Mood Time 1 | Low SES | 6.77$_a$ | 5.99$_b$ |
| | | (1.96) | (2.13) |
| | High SES | 6.83$_a$ | 6.54$_a$ |
| | | (1.69) | (1.47) |
| | Total | 6.81$_a$ | 6.26$_b$ |
| | | (1.80) | (1.85) |
| Mood Time 2 | Low SES | 6.38$_a$ | 5.48$_b$ |
| | | (1.91) | (2.18) |
| | High SES | 6.69$_a$ | 6.71$_a$ |
| | | (1.61) | (1.41) |
| | Total | 6.56$_a$ | 6.10$_b$ |
| | | (1.74) | (1.93) |
| Perceived Influence | Low SES | 2.88$_a$ | 3.19$_a$ |
| | | (1.65) | (2.35) |
| | High SES | 3.44$_a$ | 2.31$_b$ |
| | | (1.97) | (1.92) |
| | Total | 3.21$_a$ | 2.75$_b$ |
| | | (1.86) | (2.18) |
| Personal Control | Low SES | 4.89$_a$ | 3.94$_b$ |
| | | (1.14) | (1.59) |
| | High SES | 5.08$_a$ | 4.28$_b$ |
| | | (1.17) | (1.42) |
| | Total | 5.00$_a$ | 4.11$_b$ |
| | | (1.16) | (1.51) |
| Economic Risks | Low SES | -0.58$_a$ | 1.13$_b$ |
| | | (0.92) | (1.12) |
| | High SES | -0.88$_a$ | 1.07$_b$ |
| | | (0.82) | (1.09) |
| | Total | -0.75$_a$ | 1.10$_b$ |
| | | (0.87) | (1.10) |
| General Risks | Low SES | -1.32$_a$ | 0.58$_b$ |
| | | (0.95) | (1.29) |
| | High SES | -1.31$_a$ | 0.62$_b$ |
| | | (0.98) | (1.23) |
| | Total | -1.32$_a$ | 0.60$_b$ |
| | | (0.97) | (1.26) |
| Progressivity of Taxation | Low SES | 23.88$_a$ | 29.24$_b$ |
| | | (8.63) | (9.48) |
| | High SES | 28.00$_a$ | 27.06$_a$ |
| | | (12.71) | (8.07) |
| | Total | 26.32$_a$ | 28.14$_a$ |
| | | (11.38) | (8.83) |

Note: Mean values within rows that do not share the same subscript are significantly different at the $p$ = .05 level.

compared to participants assigned to the high socio-economic status and collective mind-set condition, $t(165) = -3.72$, $p < .001$.

## Main analyses

For each of our main dependent variables, we conducted a 2 (mind-set: personal, collective) x 2 (manipulated socio-economic status: low, high) analysis of variance (AVOVA). All analyses of variance reported below were conducted also including participants' mood change (i.e., mood at Time 2 minus mood at Time 1) and scores of perceived influence as covariates. None of the results changed.

**Hypothesis 1: Personal control.** We expected that participants would perceive greater control over their own future than over the nation's future. In line with Hypothesis 1, a significant effect of mind-set was found, $F(1,302) = 31.42$, $p < .001$, $\eta^2_p = .09$. Participants reported greater personal control in the personal than in the collective mind-set condition (Table 2). Neither manipulated socio-economic status nor the interaction between mind-set and manipulated socio-economic status reached statistical significance ($ps > .10$).

**Hypothesis 2a: Economic risks.** Hypothesis 2a predicted that participants would judge expected economic risks less likely for themselves but more likely for Italy. As expected, a significant effect of mind-set condition was found, $F(1, 302) = 253.64$, $p < .001$, $\eta^2_p = .46$. As shown in Table 2, participants were optimistic about their personal future, judging their future personal economic risks below average, one-sample $t(146) = -10.46$, $p < .001$; yet, they were pessimistic about the future of the nation, judging expected collective economic risks above average compared to other OCSE countries, one-sample $t(158) = 12.65$, $p < .001$. Again, neither manipulated socio-economic status nor the interaction between mind-set and manipulated socio-economic status reached statistical significance ($ps > .10$).

**Hypothesis 2b: General risks.** In line with Hypothesis 2b, a significant main effect of mind-set emerged also for expected general risks, $F(1, 302) = 214.78$, $p < .001$, $\eta^2_p = .42$. Again, participants were optimistic about their personal future (see Table 2), judging their personal general risks below average, one-sample $t(146) = -16.48$, $p < .001$, but they were pessimistic about the future of the nation, judging collective general risks above average compared to other OCSE countries, one-sample $t(158) = 6.00$, $p < .001$. No other effect was found ($ps > .85$).

**Hypothesis 3: Mediation analyses.** According to Hypothesis 3, differences in perceived control would account for the different levels of optimism at the personal vs. collective level. To test this hypothesis, a mediation analysis was conducted using PROCESS (Model 4) computational tool for conditional process analysis [98]. Scores of economic risks were used as the criterion variable in the model. Mind-set condition (0 = personal, 1 = collective) was entered as predictor, whereas personal control was modeled as centered mediator. As shown in Fig 2, mind-set condition predicted both personal control, $t = -5.75$, $p < .001$, and economic risks, $t = 14.78$, $p < .001$. Moreover, when personal control and mind-set were entered simultaneously in the model predicting economic risks, the effect of personal control was significant, $t = -2.64$, $p = .009$, indicating that higher personal control led to greater optimism about future economic risks. The CI (with 5,000 resamples) for the estimate of the indirect effect on economic risks through personal control did not include zero (95% CI [.02, .20]).

The same mediation analysis was then conducted including general risks as final outcome. Again, mind-set condition predicted both personal control, $b = -.89$, $t = -5.75$, $p < .001$, and general risks, $b = 1.96$, $t = 14.38$, $p < .001$. However, when personal control and the mind-set condition were entered simultaneously into the model predicting general risks, the effect of personal control was not significant, $b = .05$, $p > .30$, indicating that participants' personal control did not affect participants' estimates of general risks in the future.

**Study 1**

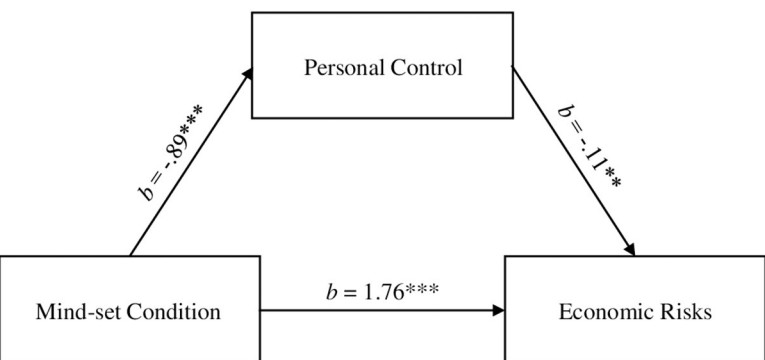

**Study 2**

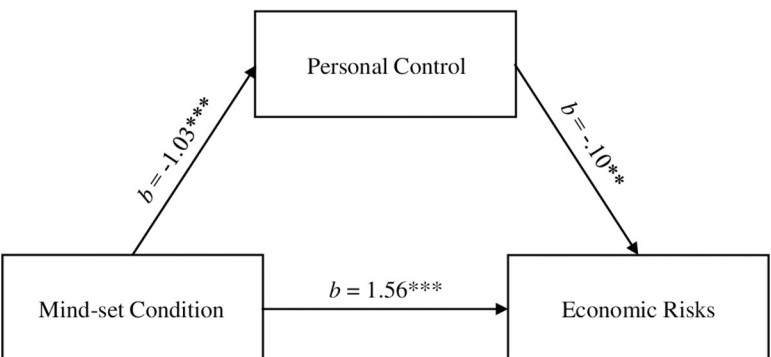

**Fig 2. Study 1 and Study 2.** Results of mediation analyses testing the indirect effects of mind-set condition (0 = personal, 1 = collective) on economic risks via personal control. Note. Study 1: $N$ = 306. Study 2: $N$ = 384. ***$p <$ .001, **$p <$ .01, *$p <$ .05.

**Hypothesis 4: Progressivity of taxation.** We had predicted that those who were made to feel poor and were in a collective mind-set would be most likely to support progressive taxation. The 2 (mind-set: personal, collective) x 2 (manipulated socio-economic status: low, high) AVOVA for progressivity of taxation revealed only the predicted interaction, $F(1, 302) = 7.37$, $p = .007$, $\eta^2_p = .02$. As shown in Table 2, personal vs. collective mind-set was irrelevant for participants in the high socio-economic status: they indeed showed similar tax preferences

regardless of whether they had been assigned to the personal or collective mind-set condition, $t(165) = .56$, $p = .57$. Conversely, mind-set condition was crucial for participants in the low socio-economic status: they tended to propose a higher progressive taxation when assigned to the collective than personal mind-set, $t(137) = -3.43$, $p = .001$. We also calculated a score of taxation using the variance of the 5 tax brackets, with higher values of this index of tax variance reflecting greater progressivity of self-generated taxation. Results remained unaltered, $F(1, 302) = 10.23$, $p = .002$, $\eta^2_p = .03$.

**Hypothesis 5: Moderated mediation analysis.** To test the fifth hypothesis, a moderated mediation model was conducted to investigate whether personal control and economic risks mediated the relation between mind-set condition and progressivity of taxation, further considering manipulated socio-economic status as a moderator of the relation between participants' perceived economic risks and scores of progressivity of self-generated taxation (PROCESS, Model 87; [98]). Therefore, progressivity of taxation was entered in the model as the criterion variable. Mind-set condition was used as predictor, whereas personal control and economic risks were modeled as centered serial mediators, respectively. Manipulated socio-economic status was included as a moderator. In line with results reported above, mind-set condition predicted personal control, $b = -.89$, $t = -5.75$, $p < .001$, and economic risks, $b = 1.76$, $t = 14.78$, $p < .001$. Personal control was a reliable predictor of economic risks, as well, $b = -.11$, $t = -2.64$, $p < .009$. Therefore, for participants in the collective, but not personal, mind-set condition personal control was reduced. Reduced personal control, in turn, increased participants' estimates of future economic risks. However, when mind-set, personal control, economic risks, manipulated socio-economic status, and the interaction between economic risks and manipulated socio-economic status were entered simultaneously in the model predicting participants' scores of progressivity of taxation, the expected interaction between economic risks and manipulated socio-economic status was only close to reaching conventional levels of significance, $b = 1.53$, $t = 1.77$, $p = .07$, $\omega = .15$; 95% CI [-.01, .43]. This result did not allow us to consider the fifth hypothesis fully confirmed.

The same moderated mediation analysis was also conducted including indexes of tax variance as final outcome. Again, mind-set condition predicted personal control, $b = -.89$, $t = -5.75$, $p < .001$, and economic risks, $b = 1.76$, $t = 14.78$, $p < .001$, and personal control was a reliable predictor of economic risks, $b = -.11$, $t = -2.64$, $p < .009$. Importantly, when mind-set, personal control, economic risks, manipulated socio-economic status, and the interaction between economic risks and manipulated socio-economic status were entered simultaneously in the model predicting indexes of tax variance, the effect of Economic Risks x Manipulated Socio-Economic Status was (almost) significant, $b = 18.91$, $t = 1.91$, $p = .056$. Noticeably, whereas the direct effect of mind-set on progressivity of taxation was not significant ($b < -6.55$, $t < -.36$, $p > .70$; [99]), bootstrap bias corrected CI (with 5000 bootstrap samples) of the overall moderated mediation index for personal control and economic risks in serial order was entirely above zero, $\omega = 1.85$; 95% CI [.06, 5.28]. Given the converging, but non-significant trend obtained for the two indices of progressivity of taxation ($p = .07$ and $p = .056$), we decided that it was premature to conclude that personal control and expected economic risks mediated the relation between personal vs. collective mind-set condition and progressivity of taxation for participants led to feel relatively poor.

## Discussion

In the present study personal control emerged as a psychological mechanism that allowed to explain the asymmetry between personal optimism and collective pessimism. As hypothesized, participants perceived greater personal control over their own future than over the nation's

future (Hypothesis 1). Participants also judged both economic and general risks in the future less likely for themselves than for others, but more likely for Italy than for other OECD countries, thus showing personal optimism, but collective pessimism (Hypothesis 2a and Hypothesis 2b).

Furthermore, in line with our hypothesis that personal control would account in large parts for the different levels of optimism in the personal vs. collective mind-set condition (Hypothesis 3), we found that personal control mediated the relation between mind-set condition and economic, but not general, risks in the future. Therefore, in the present study differences between personal optimism and collective pessimism about economic risks were explained by underlying control beliefs. However, this appeared not to be true for general risks, even though additional evidence is needed to draw firm conclusions.

Importantly, to our knowledge, for the first time this study provided tentative experimental evidence that a focus on the collective fostered support for concrete redistribution strategies, such as more progressive taxation, only among individuals who felt economically disadvantaged. Specifically, the ANOVA on indices of progressivity of taxation revealed only a significant interaction between manipulated socio-economic status and mind-set, indicating that participants in the low socio-economic status tended to propose more progressive taxation when assigned to the collective than personal mind-set condition. Conversely, no difference in tax preferences emerged between the personal and the collective mind-set condition for participants in the high socio-economic status. However, results from the moderated mediation analysis did not allow us to draw firm conclusions about the predicted role of personal control and expected economic risks in explaining the relation between personal vs. collective mind-set condition and progressivity of taxation for participants in the low socio-economic status.

One limitation of this experiment is that, to assess expected future risks, we developed a scale including 4 economic risks and only 3 general risks. Moreover, economic risks were kept identical across mind-set conditions, but framed so as to refer either to personal or collective risks. Conversely, general risks varied in content within and across mind-set conditions (e.g., divorce, alcoholism, flood). This methodological limit might represent a reasonable explanation for why in the present study a) personal control correlated more strongly with economic ($r = -.35$, $p < .001$) than with general risks ($r = -.25$, $p < .001$), and b) personal control explained the relation between mind-set condition and expected economic, but not general, risks.

Another limit of Study 1 regards the way in which the mind-set condition was operationalized. Depending on the mind-set condition to which participants had been assigned (i.e., personal vs. collective), we used three items both to trigger a focus on the self or on the collective, and to assess beliefs of personal control. Therefore, one could argue that this is both an experimental manipulation and a dependent variable. We therefore conducted a second experiment that pursued three aims. First, we aimed at overcoming the methodological limitations of Study 1. Second, we aimed at replicating the results of our first experiment, also regarding the effects of personal control on economic risks, as well as the interactive effects of mind-set and manipulated socio-economic status conditions on participants' tax preferences. Finally, we were interested in further testing the hypothesized role of personal control and economic risks in explaining the relation between mind-set condition and tax preferences for participants made to feel relatively poor (Hypothesis 5).

## Study 2

The second study was preregistered via AsPredicted (#39658) under the title "Predicting the future, Study April 2020" (available from https://aspredicted.org/blind.php?x=js8tw5; please

note that the number of the hypotheses in the text do not correspond to the number of the pre-registered ones). However, due to the Covid-19 pandemic, data collection was done different than originally planned (namely over the internet rather than in person), as will be explained below.

## Method

**Participants.** Data were collected during the period from 24th of April and 3rd of May 2020. Given the recommendations of health authorities and governmental restrictions aimed at containing and slowing down the spread of the COVID-19 virus, Study 2 was conducted totally over the internet. In exchange for course credit, students of an introductory course of Psychology e-mailed to neighbors, acquaintances, etc. an invitation to take part in a study aimed at investigating individuals' beliefs about future everyday life events. Those who agreed to participate in the study received *via* email the link to the study URL where they signed the consent form and then performed the experimental tasks. In total, 209 men and 256 women completed the on-line survey and provided their consent to use their data. As in our first experiment, the sample included only participants who (in Section 1 of the survey) indicated to have a personal monthly income ranging from 1200 to 1800 euros. Also, in line with the exclusion criteria specified a priori, 15 participants were excluded because they had provided illogical responses on the taxation task (indicating a 0 tax rate for all 5 tax brackets or indicating tax percentages that went up and down across brackets). Additional 15 respondents failed the manipulation check (i.e., did not correctly remember which place on the social ladder they had been assigned to), and 51 participants (11%) did not agree at all with the assigned position on the social ladder (personal agreement).

This resulted in a final sample of 384 participants (215 women, 169 men). Although we were originally aiming at only 300 participants (see preregistration), the type of data collection (via internet rather than in person) made it impossible to known *a priori* how many partici-pants would fall into the predefined salary range and meet all the remaining criteria. Thus, we ended up with more participants than planned and decided to maintain all of the participants who passed the pre-defined criteria and provided the permission to use their data. Participants were all residents of Southern Italy, with age ranging from 21 to 73 ($M_{age}$ = 44.18, $SD$ = 11.05). The sample mostly consisted of employees ($n$ = 341; 89%); 212 participants (55%) held a uni-versity degree, 150 (39%) a high school diploma, whereas 22 respondents (6%) had the lowest formal qualification.

## Procedure and measures

Measures and procedure were virtually identical to Study 1, with the exception of three meth-odological adjustments (i.e., mind-set manipulation, measure of expected risks, and the addi-tion of a taxation measure specifically addressing the rich). Therefore, as in Study 1, participants completed one of four surveys, depending on the condition to which they had been randomly assigned to (i.e., low socio-economic status and personal mind-set, low socio-economic status and collective mind-set, high socio-economic status and personal mind-set, high socio-economic status and collective mind-set). In Section 1 of the survey, respondents rated their current mood and provided the same socio-demographic information as in our first experiment. The manipulation of socio-economic status was also identical to Study1.

The mind-set manipulation was different from that employed in Study 1. At the beginning of Section 2, participants' mind-set was manipulated in the following way. Respondents were invited to concentrate their attention and thoughts either "on themselves and their lives" (per-sonal mind-set) or "on Italy and its citizens" (collective mind-set), depending on the condition.

To help participants with the task, instructions were accompanied by a red stylized picture of a person (personal mind-set condition), or a map of Italy including stylized persons (collective mind-set condition), positioned in the center of the screen. Afterward, as in Study 1, respondents completed measures of personal control (Cronbach's $\alpha$ = .75) and perceived influence of nation on self.

Expected risks were then assessed using a brief scale including 8 hypothetical events that differed from that of Study 1. Four events involved economic risks (i.e., "capacity to deal with economic problems", reversed scoring; "needing a loan to deal with the increased cost-of-living"; "having a very good salary", reversed scoring; "unemployment") and 4 events were general risks (i.e., "respiratory problems due to air pollution"; "car accident"; "drinking problems"; "heart attack"). These events were identical across conditions but framed so as to refer either to the individual participant (personal mind-set condition) or to Italy (collective mind-set condition). For instance, in the personal condition the item dealing with unemployment read "likelihood of that you will remain unemployed (losing the job or getting fired)", whereas in the collective condition was the following: "likelihood that Italy will see an increase in unemployment". For each item, participants either indicated the probability that the event could happen to them personally in the next twenty years, compared to the average of persons of the same age and sex (personal mind-set condition) or they indicated the probability that the event could happen to Italy in the next twenty years, compared to the average of world nations with similar features, such as geographical size, average age and educational level of the population (collective mindset condition). Comparative likelihood ratings were provided on scales ranging from -3 (*absolutely below average*) to +3 (*absolutely above average*), with 0 as the midpoint (*within the average*). Therefore, participants either rated each risk for themselves compared to similar peers, or for Italy compared to similar countries. As in Study 1, single indexes of economic risks (Cronbach's $\alpha$ = .70) and general risks (Cronbach's $\alpha$ = .64) were calculated such that negative scores indicate optimism and positive scores pessimism about the future.

In Section 3 of the survey, participants performed the same taxation task as in Study 1. After completing the task, in the present experiment respondents were also asked to indicate whether the government should reduce (coded -1), leave unchanged (coded 0), or increase (coded +1) two taxes: (a) The personal income tax for extremely wealthy people, and (b) the tax on luxury goods. Participants' responses to these two items were summed into a single score of taxing the rich (Cronbach's $\alpha$ = .67), which ranged from -2 (reducing both taxes for the rich) to +2 (increasing both taxes for the rich). This brief task was included solely for an explorative purpose. Specifically, we were interested in testing an alternative measure of support for concrete redistribution strategies.

Finally, as in Study 1, respondents completed the manipulation check and rated their personal agreement with the assigned socio-economic status (personal agreement). Prior to dismissal, participants were debriefed, thanked for their participation, and given the opportunity to either withdraw their data or sign a release form. Thirty-seven participants decided to withdraw their data.

## Results

### Preliminary analyses

To check the success of our manipulation of participants' socio-economic status, an ANOVA was conducted on personal agreement, using manipulated socio-economic status (low, high) as independent variable. As in Study 1, average agreement with the assigned position did not reliably differ between low ($M$ = 1.81, $SD$ = .78) and high socio-economic status condition ($M$ = 1.96, $SD$ = .78; $F$ = 3.06, $p$ > .08). Moreover, participants resulted approximately equally

**Table 3. Study 2 (*N* = 384).** Zero-order correlations among study variables (Family Size, Number of Minors in the Family, Number of People Contributing to the Family Income, Mood Time 1, Mood Time 2, Perceived Influence, Personal Control, Economic Risks, General Risks, Progressivity of Taxation, Taxing the Rich).

| Variables | Correlations | | | | | | | | | | |
|---|---|---|---|---|---|---|---|---|---|---|---|
| | 1 | 2 | 3 | 4 | 5 | 6 | 7 | 8 | 9 | 10 | 11 |
| 1. Family Sizie | – | | | | | | | | | | |
| 2. N° of Minors | .439*** | – | | | | | | | | | |
| 3. N° People Contributing | .383*** | -.028 | – | | | | | | | | |
| 4. Mood Time 1 | .059 | .053 | -.130** | – | | | | | | | |
| 5. Mood Time 2 | .035 | .052 | -.109* | .844*** | – | | | | | | |
| 6. Perceived Influence | .067 | .085 | -.012 | -.105* | -.121* | – | – | | | | |
| 7. Personal Control | -.042 | -.011 | .041 | .107* | .113* | -.010 | | | | | |
| 8. Economic Risks | -.008 | .044 | -.022 | -.100* | -.122* | .225*** | -.352*** | – | – | | |
| 9. General Risks | -.056 | -.024 | -.067 | -.077 | -.125* | .158** | -.254** | .582*** | | | |
| 10. Progressivity of Taxation | -.012 | .007 | -.017 | .044 | -.002 | .015 | -.050 | .058 | .022 | – | |
| 11. Taxing the Rich | .061 | .067 | .078 | -.024 | -.085 | -.071 | -.003 | -.006 | -.043 | .387*** | – |

***$p < .001$

**$p < .01$

*$p < .05$.

distributed across conditions (low socio-economic status and personal mind-set, high socio-economic status and personal mind-set, low socio-economic status and collective mind-set, high socio-economic status and collective mind-set) in terms of gender, age, education, and occupation (all *p*s > .08).

Table 3 shows zero-order correlations among participants' family characteristics (family size, number of family members below 18 years of age, number of people contributing to the family income) and main study variables (mood Time 1, mood Time 2, perceived influence, personal control, economic risks, general risks, progressivity of taxation, taxing the rich) As in Study 1, participants' family size correlated positively with the number of family members below 18 years of age, as well as with the number of people contributing to the family income. Again, no relation emerged among these variables nor with personal control, economic risks, general risks, progressivity of taxation, taxing the rich, or personal agreement (all *p*s > .10). As in Study 1, greater personal control was associated with higher optimism about both economic and general risks.

In Table 4 are presented descriptive statistics for mood Time 1, mood Time 2, perceived influence, personal control, economic risks, general risks, progressivity of taxation, and taxing the rich across experimental conditions (manipulated socio-economic status, mind-set).

As in Study 1, we tested for the effects of participants' sociodemographic characteristics on our main study variables. Multiple regression analyses were conducted on personal control, economic risks, general risks, and progressivity of taxation. Manipulated socio-economic status (low = 0, high = 1), mind-set (personal = 0, collective = 1), gender (male = 0, female = 1), age, education, occupation, and their interactions (second block) were entered as predictors. In the model predicting participants' scores of personal control, the effect of Gender x Mindset was significant, $b = .25$, $t(371) = 2.93$, $p = .004$, indicating that men perceived greater personal control than women in the personal mind-set (men: $M = 5.6$, $SD = 0.93$; women: $M = 5.3$, $SD = 0.87$; $t(196) = 2.78$, $p = .006$), whereas women felt more personal control than men in the collective mind-set (men: $M = 4.2$, $SD = 1.49$; women: $M = 4.6$, $SD = 1.48$; $t(184) = -1.95$, $p = .05$). However, both men and women reported greater personal control in the personal than in the collective mind-set condition (men: $t(167) = 7.60$, $p < .001$; women: $t(213) =$

**Table 4. Study 2 ($N$ = 384).** Means and standard deviations of Mood Time 1, Mood Time 2, Perceived Influence, Personal Control, Economic Risks, General Risks, Progressivity of Taxation, and Taxing the Rich as a function of Manipulated Socio-Economic Status (Low SES, High SES) and participants' Mind-Set (Personal, Collective).

| | | Mind-Set | |
|---|---|---|---|
| **Variables** | | **Personal** | **Collective** |
| Mood Time 1 | Low SES | 6.23[a] | 5.99[a] |
| | | (1.99) | (1.77) |
| | High SES | 6.29[a] | 6.32[a] |
| | | (1.81) | (1.55) |
| | Total | 6.26[a] | 6.16[a] |
| | | (1.90) | (1.66) |
| Mood Time 2 | Low SES | 6.07[a] | 5.61[a] |
| | | (2.05) | (1.82) |
| | High SES | 6.31[a] | 6.21[a] |
| | | (1.64) | (1.63) |
| | Total | 6.20[a] | 5.91[a] |
| | | (1.85) | (1.75) |
| Perceived Influence | Low SES | 5.31[a] | 5.84[b] |
| | | (1.76) | (1.65) |
| | High SES | 5.09[a] | 5.87[b] |
| | | (1.70) | (1.65) |
| | Total | 5.20[a] | 5.85[b] |
| | | (1.73) | (1.65) |
| Personal Control | Low SES | 5.30[a] | 4.44[b] |
| | | (0.90) | (1.49) |
| | High SES | 5.52[a] | 4.33[b] |
| | | (0.91) | (1.50) |
| | Total | 5.41[a] | 4.38[b] |
| | | (0.91) | (1.49) |
| Economic Risks | Low SES | -0.54[a] | 0.99[b] |
| | | (0.95) | (0.92) |
| | High SES | -0.74[a] | 1.06[b] |
| | | (0.82) | (1.00) |
| | Total | -0.64[a] | 1.02[b] |
| | | (0.89) | (0.96) |
| General Risks | Low SES | -0.51[a] | 0.74[b] |
| | | (0.83) | (0.87) |
| | High SES | -0.61[a] | 0.69[b] |
| | | (0.79) | (0.89) |
| | Total | -0.56[a] | 0.71[b] |
| | | (0.81) | (0.88) |
| Progressivity of Taxation | Low SES | 25.51[a] | 29.65[b] |
| | | (8.22) | (9.59) |
| | High SES | 27.26[a] | 26.99[a] |
| | | (7.60) | (10.41) |
| | Total | 26.41[a] | 28.31[b] |
| | | (7.24) | (10.56) |
| Taxing the Rich | Low SES | 0.74[a] | 0.85[a] |
| | | (1.36) | (1.21) |

(*Continued*)

**Table 4.** (Continued)

| Variables | | Mind-Set | |
|---|---|---|---|
| | | Personal | Collective |
| | High SES | 0.79$_a$ | 0.76$_a$ |
| | | (1.27) | (1.40) |
| | Total | 0.77$_a$ | 0.80$_a$ |
| | | (1.31) | (1.31) |

Note: Mean values within rows that do not share the same subscript are significantly different at the $p$ = .05 level.

4.18, $p$ < .001), thus suggesting that the manipulation was effective for both genders. Importantly, no other interaction effect was found ($ps$ > .10). Therefore, we will not discuss sociodemographic variables further.

**Mood.** Mood scores were the dependent variable of a 2 (mind-set: personal, collective) x 2 (manipulated socio-economic status: low, high) x 2 (mood: Time 1, Time 2) repeated measures ANOVA with the last variable within-participants. Results revealed a main effect of mood, $F(1, 380)$ = 9.38, $p$ = .002, $\eta^2_p$ = .02, indicating that participants were in a better mood at the beginning (Time 1: $M$ = 6.21, $SD$ = 1.78) than at the end of the study (Time 2: $M$ = 6.06, $SD$ = 1.81). Results also showed a significant Mood x Manipulated Socio-Economic Status interaction effect, $F(1, 380)$ = 4.89, $p$ = .03, $\eta^2_p$ = .01: for participants assigned to the low socio-economic status mood at Time 2 was lower ($M$ = 5.8, $SD$ = 1.95) than mood at Time 1 ($M$ = 6.11, $SD$ = 1.88), $t(187)$ = 3.69, $p$ < .001. Conversely, no change in mood was found for participants in the high socio-economic status (Time 1: $M$ = 6.31, $SD$ = 1.68; Time 2: $M$ = 6.27, $SD$ = 1.63; $t(195)$ = .57, $p$ = .57).

**Perceived influence of nation on self.** Participants' scores of perceived influence were subjected to a one-way ANOVA, using mind-set (personal, collective) and manipulated socio-economic status (low, high) condition as independent variables. A significant effect of mind-set was found, $F(1, 380)$ = 14.33, $p$ < .001, $\eta^2_p$ = .04: in the collective mind-set participants felt personally more affected by what happens in their nation, as compared to the personal mind-set condition (see Table 4).

## Main analyses

We conducted a series of five ANOVAs on participants' scores of personal control, economic risks, general risks, progressivity of taxation, and taxing the rich, again using mind-set (personal, collective) and manipulated socio-economic status condition as the independent variables for each. Again, all results reported below remained unaltered even when participants' mood change (i.e., mood at Time 2 minus mood at Time 1) and perceived influence were included as covariates.

**Hypothesis 1: Personal control.** As predicted in Hypothesis 1, participants reported greater personal control in the personal than in the collective mind-set condition (see Table 4), $F(1, 380)$ = 66.71, $p$ < .001, $\eta^2_p$ = .15. No other effect emerged ($ps$ > .20).

**Hypothesis 2a: Economic risks.** As predicted in Hypothesis 2a, a main effect of mind-set emerged, $F(1, 380)$ = 311.38, $p$ < .001, $\eta^2_p$ = .45. As shown in Table 4, in line with results of Study 1, participants were optimistic in the personal mind-set condition, thus envisaging their risk of experiencing negative economic events lower than the average risk of their peers, one-sample $t(197)$ = -10.17, $p$ < .001, but they were pessimistic in the collective mind-set condition, judging their nation's risk higher than the average risk of similar nations, one-sample $t(185)$ = 14.56, $p$ < .001. No other effect was found ($ps$ > .15).

**Hypothesis 2b: General risks.**   Similar to Study 1, a significant main effect of mind-set emerged also for expected general risks, $F(1, 380) = 217.84$, $p < .001$, $\eta^2_p = .36$. Again, in line with Hypothesis 2b, participants (see Table 4) showed personal optimism, judging their personal general risks below average, one-sample $t(197) = -9.75$, $p < .001$, but collective pessimism, judging their nation's general risks above average, one-sample $t(185) = 11.09$, $p < .001$. No other effect was found ($ps > .85$).

**Hypothesis 3: Mediation analyses.**   To test Hypothesis 3, a first mediation analysis was conducted (PROCESS, Model 4; [98]) using scores of economic risks as the criterion variable, mind-set condition (0 = personal, 1 = collective) as predictor, and personal control as centered mediator. As shown in Fig 2, mind-set condition predicted personal control, $t = -8.20$, $p < .001$, and economic risks, $t = 15.39$, $p < .001$. Moreover, in line with results from Study 1, when personal control and mind-set were entered simultaneously in the model predicting economic risks, the effect of personal control was significant, $t = -2.68$, $p = .008$, thus showing that higher personal control led to greater optimism about future economic risks. The CI (with 5,000 resamples) for the estimate of the indirect effect on economic risks through personal control did not include zero (95% CI [.02, .21]).

The same mediation analysis was then conducted including general risks as final outcome. Again, mind-set condition predicted both personal control, $b = -1.03$, $t = -8.20$, $p < .001$, and general risks, $b = 1.26$, $t = 13.42$, $p < .001$. However, when personal control and the mind-set condition were entered simultaneously into the model predicting general risks, the effect of personal control was not significant, $b = -.02$, $p > .58$. Therefore, as in Study 1, participants' personal control did not affect participants' perception of future general risks.

**Hypothesis 4: Progressivity of taxation.**   As shown in Table 4, participants tended to propose a lower progressive taxation in the personal than the collective mind-set condition, $F(1, 380) = 4.18$, $p = .04$, $\eta^2_p = .01$. Importantly, in line with results of Study 1, a Manipulated Socio-Economic Status x Mind-Set interaction effect was found, $F(1, 380) = 5.46$, $p = .020$, $\eta^2_p = .02$. Again, participants made to feel relatively well-off showed similar tax preferences regardless of whether they had been assigned to the personal or collective mind-set condition, $t(194) = .21$, $p = .83$. Conversely, participants made to feel relatively poor proposed a higher progressive taxation when in the collective than personal mind-set condition, $t(186) = -3.00$, $p = .003$. As in Study 1, we also calculated an index of tax variance, with higher values reflecting greater progressivity of self-generated taxation. Results remained unaltered, $F(1, 302) = 10.23$, $p = .002$, $\eta^2_p = .03$.

The same analysis for the two-item taxing the rich measure showed that neither mind-set, nor manipulated socio-economic status, or the interaction between mind-set and manipulated socio-economic status reached statistical significance ($ps > .58$). Regardless of condition, participants were in favor of increasing taxes for the rich ($M = .78$, $SD = 1.31$), which exceeded the scale midpoint, $t(383) = 11.75$, $p < .001$.

**Hypothesis 5: Moderated mediation analyses.**   A moderated mediation model (PROCESS, Model 87; [98]) was conducted to test Hypothesis 5. Participants' scores of progressivity of taxation were entered in the model as the criterion variable. Mind-set condition was included as predictor, whereas personal control and economic risks were modeled as centered serial mediators respectively. Manipulated socio-economic status was used as a moderator. Results are presented in Fig 3. Mind-set condition predicted personal control, $t = -8.20$, $p < .001$, and economic risks, $t = 15.39$, $p < .001$. Personal control predicted economic risks, as well, $t = -2.68$, $p < .008$. Therefore, as in Study 1, for participants in the collective, but not personal, mind-set condition personal control was reduced. Reduced personal control, in turn, increased participants' estimates of future economic risks. Crucially, this time, when mind-set, personal control, economic risks, manipulated socio-economic status, and the interaction

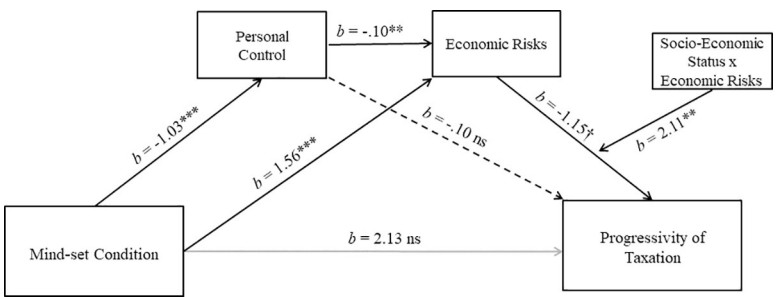

**Fig 3. Study 2.** Results of moderated mediation analysis testing the indirect effects of mind-set condition (0 = personal; 1 = collective) on progressivity of taxation via personal control and economic risks (as serial mediators). Note. $N$ = 384. *** $p < .001$, ** $p < .01$, * $p < .05$, †$p < .07$.

between economic risks and manipulated socio-economic status were entered simultaneously in the model predicting participants' scores of progressivity of taxation, the effect of Economic Risks x Manipulated Socio-Economic Status was significant, $t$ = 2.76, $p$ = .006, indicating that for participants in the low, but not high, socio-economic status, greater expected economic risks led to higher progressivity of taxation. Noticeably, bootstrap bias corrected CI (with 5000 bootstrap samples) of the overall moderated mediation index for personal control and economic risks in serial order was entirely above zero, ω = .22; 95% CI [.02, .55]. Therefore, for participants in the low, but not high, socio-economic status, personal control and expected economic risks mediated the relation between personal vs. collective mind-set condition and progressivity of taxation.

As in Study 1, the same moderated mediation analysis was then conducted including indexes of tax variance as final outcome. Again, mind-set condition predicted personal control, $b$ = -1.03, $t$ = -8.20, $p < .001$, and economic risks, $b$ = 1.56, $t$ = 15.39, $p < .001$. Personal control predicted economic risks, as well, $b$ = -.10, $t$ = -2.68, $p < .008$. Importantly, when mind-set, personal control, economic risks, manipulated socio-economic status, and the interaction between economic risks and manipulated socio-economic status were entered simultaneously in the model predicting indexes of tax variance, the effect of Economic Risks x Manipulated Socio-Economic Status was significant, $b$ = 16.88, $t$ = 2.06, $p$ = .04. However, bootstrap bias corrected CI (with 5000 bootstrapping samples) of the overall moderated mediation index included zero, ω = 1.77; 95% CI [-.08, 4.78].

**Progressivity of taxation: A pooled data analysis.** Although both Study 1 and Study 2 confirmed the predicted interaction between manipulated socio-economic status and mind-set on support for tax-based redistribution (Hypothesis 4), the result pattern was slightly different. In both studies, those who felt relatively well-off (high socio-economic status) were unaffected by the mind-set manipulation. However, the effect of mind-set on those who were made to feel disadvantaged differed across studies: participants in the low socio-economic status reduced their support for progressivity of taxation when in a personal mind-set in Study 1 but increased their support for redistribution when in a collective mind-set in Study 2.

To understand whether personal mind-set had reduced or whether collective mind-set had increased support for redistribution among the relatively poor participants, or whether both processes had operated to a similar extent, we conducted a pooled data analysis. A 2 (manipulated socio-economic status: low, high) x 2 (mind-set: personal, collective) x 2 (study: study 1, study 2) ANOVA was conducted on participants' scores of progressivity of taxation. Results revealed a significant interaction between mind-set and manipulated socio-economic status, $F$ (1, 682) = 13.10, $p < .001$, $\eta^2_p$ = .02, which, importantly, was not moderated by study (study 1 vs. study 2), $F(1, 682)$ = .50, $p$ = .48, $\eta^2_p$ = .001. In the high socio-economic status progressive

taxation was very similar, regardless of mind-set condition (personal: $M = 27.60$, $SD = 10.25$; collective: $M = 27.02$, $SD = 9.38$). Compared to participants who felt relatively rich, conversely, those who felt relatively poor proposed a less progressive taxation when in a personal mind-set ($M = 24.88$, $SD = 8.39$; $t(343) = 2.66$, $p = .008$), but a more progressive taxation when in a collective mind-set ($M = 29.52$, $SD = 10.07$; $t(343) = -2.33$, $p = .02$. Therefore, depending on the mind-set condition, redistribution attitudes of those who were made to feel relatively poor, but not rich, were greatly polarized.

## General discussion

Although most citizens desire greater economic equality in their countries and would personally benefit from strategies aimed at transferring wealth from richer to poorer members of society, they often fail to support concrete redistribution policies to achieve this goal [12, 13]. Over the last years, the increasing levels of wealth and income inequalities in practically all developed nations [100] and their detrimental effects on society [101] have led many social scientists to draw attention to a number of psychological factors that contribute to inhibiting support for policies aimed at redressing economic inequality [19, 27, 29, 42]. With the aim of contributing to this growing body of research, in the present work we decided to change perspective and to address possible conditions that may facilitate rather than hamper the demand of redistribution.

We therefore proposed and tested a model according to which people will support concrete redistribution policies if two conditions are met: a) They are dissatisfied with their current personal economic situation, regardless of their objective socio-economic status, and b) they take a pessimistic economic outlook at the collective rather than personal level. Two distinct lines of research stimulated and then converged on our reasoning. According to the first, demand for social change is more likely when people feel relatively disadvantaged [51]. The second line of research suggests that change (redistribution, in our case) requires the awareness that things–in the absence of action–will get worse [55, 57]. Typically, this kind of pessimistic outlook occurs when individuals think about the collective's future (e.g., the fate of their nation or of the world), but not when they envisage their own personal future [76]. People indeed tend to be overly positive and optimistic about their personal future [65], because, they feel, it is under their own control [86]. Drawing from this evidence, we developed a model according to which a collective mind-set should foster a pessimistic outlook, which, in turn, should result in concrete demand for redistribution, but only when individuals feel economically disadvantaged.

Given its complexity, we broke our model down into four distinct hypotheses, which received rather coherent support across two studies. First, in both studies we found that people, quite logically, believe to have greater control over their personal future than over the future of their nation. Second, in line with prior literature [76, 77], participants felt overly optimistic about their personal future, whereas they felt pessimistic about their collective future, as a nation. Specifically, our results show that comparing themselves or their nation to similar other individuals or nations, participants rated possible economic and general risks in the future as less likely for themselves, but as more likely for Italy. Third, across two studies we found that the asymmetry between personal optimism and collective pessimism regarding economic risks was driven by differences in perceived personal control over events of a personal and a collective character. This finding extends prior work on the role of perceived personal control over events in the unrealistic optimism bias [64, 87] by demonstrating that personal control is a psychological mechanism that can help not only to explain differences between cultures [87] and between types of possible events of a personal character [88], but also to

account for the striking contrast in risk perception between events at the personal and collective level.

Turning to our primary dependent variable, namely support for redistribution policies aimed at redressing wealth and income inequalities, our hypothesis was that participants' demand for redistribution through progressive taxation would be greater if they were in a collective mind-set *and* felt relatively economically disadvantaged. The first finding in line with this prediction comes from the ANOVAs on progressivity of taxation. In both studies results revealed a significant interaction between manipulated socio-economic status and mind-set, showing that a personal vs. collective mind-set was irrelevant for participants led to feel relatively well-off. Conversely, mind-set played an important role for those led to feel economically disadvantaged: only participants in the low socio-economic status condition indeed proposed higher progressive taxation when in a collective than in a personal mind-set (see also pooled data analysis). This finding clearly demonstrates that request for redistribution is highest when people feel poor and reason collectively about future.

The second proof comes from the moderated mediation model. Here, greater pessimism about future collective economic risks produced support for progressive taxation only among participants in the low, but not high, socio-economic status condition. Although results were stronger for variance than for difference scores of progressivity of taxation in Study 1, whereas the opposite pattern emerged in Study 2, overall, the four moderated mediation analyses yielded very similar results. Despite the complexity of our model, the two studies together provide coherent support for the idea that collective pessimism translates into demand for redistribution only when people feel relatively poor.

Results from the present work are novel also because they allow for causal inferences. Indeed, different from archival research using large survey data [56, 57], in our studies, the primary predictor variables, namely participants' socio-economic status and mind-set, were manipulated rather than assessed. Importantly, although participants were objectively in the same income range, subjective socio-economic status was successfully varied by providing them a fictitious feedback about their relative standing in the social and economic ladder. This manipulation allows us to conclude that potentially confounding variables, which typically covary with individuals' actual socio-economic status (e.g., political orientation, tax attitudes, system justification) can not account for our findings.

Although we obviously do not claim that these are the only determinants of attitudes toward redistribution, in our studies citizens' support for policies aimed at redressing economic inequality depended on the unique combination of a collective mind-set and feelings of relative personal economic deprivation. This evidence becomes especially important when viewed in the light of recent results showing that economic inequality creates a competitive social climate and fosters individualism at the expense of common fate and collective goals [45]. It is also worth noting that the overly optimistic representation of upward social mobility often conveyed by media is likely to contribute to a focus on the self and to increasing the illusory idea of a rosy personal future even in the presence of collectively bleak times [102, 103].

The present research may also have implications from an applied point of view. Our findings show that people were unlikely to support redistribution if they had an overly optimistic vision of the economic future, which is exactly what happens when they think about their personal (rather than collective) risks (see optimism bias; [64]). Therefore, it seems clear that any communication intended to promote support for redistribution policies in the population will be more effective in creating positive engagement by shifting people's attention from the personal to the collective level. In this way, besides promoting the idea that things may get worse and hence that change is needed, it may become also easier to affect beliefs about future risks faced by the community at large. In fact, whereas personal optimism is very stable over time,

the collective pessimism fluctuates greatly and is susceptible to external factors such as media coverage [90].

## Limits and future development

Given that this is the first research testing the combined role of a collective mind-set and subjective social status in attitudes towards redistribution, it is not surprising that a number of questions remain unanswered.

First of all, our results can provide only preliminary information about the mediating role of personal control. On the one side, it remains unclear why control beliefs mediated the relation between mind-set condition and expected economic, but not general, risks despite the fact that participants perceived greater control over personal than over collective risks in both cases. On the other side, even in the case of economic risks, we can not exclude the possibility that additional mediators may have played a role. Drawing on literature, perceived control seemed to be the most plausible psychological mechanism that could explain the asymmetry between personal optimism and collective pessimism [64, 86–89]. However, many mechanisms contribute to producing personal optimism [84] and, therefore, an exhaustive analysis aimed at identifying the true mediator would have to rely on diagnostic tests of mediation against other candidates that we did not assess in the present work. For example, individuals are more optimistic about their personal than their collective future also because they feel it is under their own personal responsibility [89]. It is possible that this and possibly additional psychological processes may have contributed to the relation between mind-set and expected economic and general risks above and beyond personal control. Therefore, we deem it necessary for future research to extend the present findings investigating other psychological mechanisms in addition to personal control.

Second, the driving idea of our work was that people will support redistribution policies if they take a pessimistic economic outlook at the collective level and if they feel relatively poor. To test our model, we manipulated participants' socio-economic status (high, low) and mindset (personal, collective). However, one could argue that, in the absence of a no-mind-set control condition, our experimental design does not allow to disentangle with certainty whether it is collective pessimism that makes relatively poor participants more prone to support redistribution policies, or whether personal optimism inhibits their support for redistribution. Although we acknowledge this limit, a comparison with those who felt relatively rich (see pooled data analysis) suggests that both processes may be operating, given that those who felt poor showed less support for progressive taxation than the "rich" counterparts in the collective, but more support in the personal mind-set condition. Thus, support for progressive taxation of participants who were made to feel relatively poor were greatly polarized. On the one side, collective pessimism motivated those who felt poor to endorse concrete redistribution strategies. On the other side, although not originally hypothesized, personal optimism hampered their wish for tax-based redistribution. In line with prior theorizing [18], it is plausible to argue that those who felt relatively poor and were considering their personal (rather than collective) risks, either made stronger internal attributions or justified the system more, resulting in less support for redistribution. It is up to future research to investigate the exact underlying reasons of this polarized view on progressive taxation among the poor. Another extension of our collective vs. personal mind-set manipulation may be to manipulate the optimistic or pessimistic outlook directly, for instance by informing participants that things will (or will not) get worse collectively (or personally). However, such information-based manipulations are often insufficient to induce changes in tax preferences [25]. It therefore remains to be seen whether information-based interventions work as well or better than the indirect approach

chosen here, in which people generate their own (optimistic or pessimistic) outlook. Our work, as well as literature, suggests that it is the focus on the collective *per se* that makes people pessimistic about the future, and it is the focus on the self *per se* that makes people unrealistically optimistic about the future, thus increasing and reducing, respectively, support for redistribution among the relatively poor. A systematic comparison of the two approaches may be useful both from a theoretical and applied point of view.

Third, although economists recommend a wide variety of strategies to reduce wealth and income inequalities, including income regulations (such as maximum-wage and living wage), social welfare programs, and public service policies, to measure participants support for redistribution in the present work we focused specifically on progressive taxation of personal income. Our choice was driven by the fact that progressive taxation has received increasing attention in recent years [57, 104] and has been shown to have tangible effects on people's well-being and happiness [105]. Moreover, among the various existing forms of progressive taxation (e.g., income, property, consumption), income tax is the one that citizens are most familiar with. Solely for explorative purposes, in Study 2 we included, as an additional measure of support for redistribution, attitudes towards two types of taxes specifically targeting top earners (personal income and luxury goods). However, no significant effect emerged of either manipulated socio-economic status or mind-set on this variable, possibly because participants had overall rather positive attitudes towards this type of taxation. Therefore, to fully grasp the generality of the phenomena investigated here, we encourage future research to extend our model to other types of taxes (e.g., inheritance tax that is particularly suitable for increasing social mobility), as well as to other forms of redistribution.

Fourth, our studies were conducted in Italy, a country that fares about average in economic performance among the OECD economies. For instance, it is close to the OECD average and to the OECD Europe average in GDP [106]. These features made Italy an ideal nation for our within-OECD comparison developed to assess collective economic and general risks in Study 1. Although in Study 2 participants assigned to the collective mind-set condition compared Italy to other countries with similar geographical and social characteristics, overall, the two experiments yielded very similar results, thus suggesting that the specific comparison was not essential. Nonetheless, future studies should investigate the cross-cultural generality of the model proposed here.

Finally, the fact that our second study was run during one of the peaks of the Covid-19 pandemic may have affected our results in unknown ways. Many researchers have observed "anomalous" responses to surveys administered during the pandemic, so this possibility can not be excluded with certainty. In particular, participants may have experienced worse mood and less control over their lives during the pandemic. Also, the very issue of economic inequality may have become less central due to the overwhelming health concerns that took center stage in the media (and in people's minds) at that point in time. Comparing pre-experimental mood across studies, we observe a slightly better mood among participants of (pre-Covid) Study 1 ($M = 6.53$, $SD = 1.84$) than among those of Study 2 ($M = 6.21$, $SD = 1.78$), although differences were small considering the 1–10 scale on which mood was assessed. However, participants in Study 2 reported a somewhat greater sense of control ($M = 4.91$, $SD = 1.33$) than those in (pre-Covid) study 1 ($M = 4.54$, $SD = 1.42$), Thus, there is no consistent pattern that would suggest that responses in Study 2 may have been negatively impacted by the massive disruption experienced during Covid-19. More importantly, the similarity of result patterns obtained in study 1 (pre-Covid-19) and Study 2 (during Covid-19) for all hypotheses, including the complex moderated mediation model, argues against a reliable interference of Covid-19 in this set of studies.

To conclude, according to the conceptualization of human beings as *homo economicus*, guided by rationality and self-interest, making people (especially those who are relatively poor)

think about their own economic future should motivate them to seek redistribution from which they personally would benefit. In contrast, the model presented here suggests that this strategy would make people overly optimistic about future, which in turn would hamper their desire for redistribution. Ironically, the focus on one's personal future may reduce rather than increase citizens' motivation to seek greater equality. Thus, the quest for redistribution requires the "pessimism of the intellect" without which the need for change would not become apparent. Or as stated by Gramsci in a letter to one of his brothers in December 1929: "I am pessimistic in reasoning, but optimistic in my will. In any circumstance, I consider the worst-case scenario, to set all reserves of will in motion and be able to break down the obstacle".

## Supporting information

**S1 Fig. Scales used in study 1 and study 2 to exclude from the sample participants with a personal monthly income below 1200 and above 1800 euros and to introduce the manipulation of participants' socio-economic status.**
(TIF)

**S1 File. Study 1.** Scale used to assess participants' expected risks.
(PDF)

**S1 Dataset. Dataset study 1.**
(SAV)

**S2 Dataset. Dataset study 2.**
(SAV)

## Author Contributions

**Conceptualization:** Silvia Galdi, Anne Maass.

**Data curation:** Silvia Galdi.

**Formal analysis:** Silvia Galdi, Anne Maass.

**Investigation:** Silvia Galdi, Annalisa Robbiani.

**Methodology:** Silvia Galdi, Anne Maass.

**Project administration:** Silvia Galdi, Anne Maass.

**Writing – original draft:** Silvia Galdi.

**Writing – review & editing:** Silvia Galdi, Anne Maass.

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
