## [Decision Letter · Decision Letter 0]

21 Oct 2020

PONE-D-20-23794

The bright side of pessimism: Promoting wealth redistribution under (felt) economic hardship

PLOS ONE

Dear Dr. Galdi,

Thank you for submitting your manuscript to PLOS ONE. After careful consideration, we feel that it has merit but does not fully meet PLOS ONE’s publication criteria as it currently stands. Therefore, we invite you to submit a revised version of the manuscript that addresses the points raised during the review process. In particular, both referees consider that the general motivation of the paper and the presentation of the results can be improved substantially.

We look forward to receiving your revised manuscript.

Kind regards,

Luis M. Miller, Ph.D.

Academic Editor

PLOS ONE

2. Please change "female” or "male" to "woman” or "man" as appropriate, when used as a noun.

Reviewers' comments:

Reviewer's Responses to Questions

**Comments to the Author**

1. Is the manuscript technically sound, and do the data support the conclusions?

Reviewer #1: Yes

Reviewer #2: Partly

2. Has the statistical analysis been performed appropriately and rigorously? 

Reviewer #1: Yes

Reviewer #2: No

3. Have the authors made all data underlying the findings in their manuscript fully available?

Reviewer #1: No

Reviewer #2: Yes

4. Is the manuscript presented in an intelligible fashion and written in standard English?

Reviewer #1: Yes

Reviewer #2: No

5. Review Comments to the Author

Reviewer #1: I have enjoyed reading the manuscript “The Bright Side of Pessimism: Promoting wealth redistribution under felt economic hardship”. From my perspective, this paper deals with a relevant and timely topic: how to predict attitudes towards redistribution. Moreover, it adds to the literature by presenting experimental evidence—which is important because, as the authors have argued, most studies on this topic use correlational design.

In short, I have a positive evaluation of this paper. But I also think there are still some issues that will be good to address. I will list them below.

OVERALL RATIONALE OF THE MANUSCRIPT

I think this paper does an important contribution when examining the effects on the redistribution measure. Thus, I think it will be better to focus the rationale of the manuscript—a bit more—around this effect. The differences between personal optimism and collective pessimism, and how this relates with sense of control, are interesting research questions, but they have already been examined in the literature.

Hence, I think it will be better to focus more the introduction around the redistribution research question and maybe put a bit aside the other questions—although it will still be important to include them in the manuscript, of course.

Part of this is also related with how the mediations are interpreted. I think that mediations provide preliminary information about the examined processes, but it will be important to corroborate this findings using experimental methodology (see Fiedler, Harris, & Schott, 2018, JESP; Spencer, Zanna, & Fong, 2005, JPSP). Thus, I think the authors may present this limitations of the mediation analyses in the general discussion. At the same time, they could give greater importance to the main (and interaction) effects of their experimental manipulations.

WHERE IS THE MAIN DIFFERENCE?

All in all, the authors argue that participants who where in the low class/collective pessimism condition where those who supported more the progressive redistribution policy. Although this seems true in Study 2, I am not so sure this also happens in Study 1.

Specifically, in Study 1 the means show that participants in this group (M = 29.24) were very similar to those participants assigned to the high class/collective pessimism (M = 27.06, and to the high class/personal optimism (M = 28.00) condition. Thus, in Study 1, participants that tended to differ from the other three conditions were those assigned to the low class/personal optimism condition (M = 23.88)—not those assigned to the low class/collective pessimism condition.

I think this is important because, although the interaction effect is replicated in Study 2, the direction of the means may be telling a different story. The authors may check if these are really different patterns using planned contrasts.

If there are different patterns, one way of knowing which of the patterns is more consistent—if participants in the collective/low class or participants in the personal/low class are the ones driving the effect— may be to run a pooled data (or integrative data) analysis. Performing the analysis with the pooled sample may answer the question about which group is different from the others.

WHAT IS THE COLLECTIVE PESSIMISM/PERSONAL OPTIMISM MANIPULATION CHANGING?

One of my biggest doubts was about the collective pessimism vs. personal optimism manipulation. I am not sure about what this manipulation is exactly comparing. In these studies, is it collective pessimism that make participants more aware of inequalities and that is why they tend to support more redistribution? Or is it that thinking about their (seemingly positive) personal outcomes make them less prone to redistribute money?

Said otherwise, I think that using a control group will be important to know where is the effect. In fact, the authors say that participants want to redistribute more when they are deprived and when they have “Awareness that things will get worse”. Thus, if they believe that this is what is driving the effect, I think it may be a good idea to directly manipulate this process, instead of doing the collective pessimism vs. personal optimism contrast. Authors may compare a situation in which “things will go collectively worse” against a situation in which they will stay the same.

Alternatively, the authors may argue that they will not find differences when comparing “when things will go collectively worse” with “when things will stay collectively the same”, because what is important is to believe that you will be better off (or not) in the future. In this case, I think it will be important to orthogonally manipulate both variables—and perform a 2 (High and low in collective pessimism) x 2 (High and low in personal optimism) experiment.

I think that a paper with any of this two studies may be more informative. However, if the authors are not able to run the study again, it may be important to include this in the limitations of the study—that, given that the authors do not have a control group, you could not know if the effect is driven by collective pessimism or personal optimism.

COVID-19

The authors ran Study 2 during the hardest times of the COVID-19 pandemic. As a researcher doing studies about the social psychology of economic inequality, I found that most of the studies I ran during these months tend to present unusual results.

Although it is clear that it did not influence much the effects of this study, as the authors were able to replicate their finding, I was wondering whether the authors controlled or asked some questions related with the pandemic. Or if it influenced in some way participants’ responses; for instance, the collective pessimism manipulation may be stronger after the pandemic than before.

MINOR ISSUES

• Hypothesis 4 may be rewritten. It was not very clear for me that this was a “moderated mediation” hypothesis. This, of course, may not be a deviation from the preregistration as the hypothesis will be the same—and only the wording changes.

• I tried to look for the preregistration, but I was not able to do so. Please include a link to the preregistration in the paper.

Guillermo B. Willis

Universidad de Granada

Reviewer #2: This paper studies the role played by psychological processes like personal optimism and collective pessimism to explain why people fail to support redistribution strategies to redress inequality.

The authors conduct two experimental studies to test their hypothesis. In the first, they manipulate participants’ socio-economic status and mind-set. In the second, they try to replicate results from study 1 and go further in explaining the determinants of tax preferences using personal control and economic risks as main explanatory variables.

I think the idea is worthy to study, however the analyses and results are not presented in an easy way. That makes it difficult to interpret the main results and assess whether there is something missing in the analysis that may be important. Also, the comparison between analyses it is not straightforward. I would recommend using tables to present the experimental design, the descriptive statistics and the main analyses. Showing the average, standard deviation and p-values in the main text makes it difficult to follow the main argument of the paper. Also, I would recommend to add sociodemographic variables as control in the different analyses presented in the text.

Regarding the design of the experiment I am not convinced about a couple of points.

Why do the authors restrict the sample to participants with a personal monthly income between 1200 and 1800 euros? The authors should justify this decision and explain how it could affect the results.

It is difficult to think how credible the socio-economic manipulation is as all participants are equally rich or poor in their real live. The high number of participants, in study 1 and also in study 2, that do not agree with the assigned position in the economic ladder would indicate that the manipulation is not working properly and it could have some important consequences in the results. Please, discuss these two points and explain their main consequences and how they may limit the scope of the results.

6. PLOS authors have the option to publish the peer review history of their article (what does this mean?). If published, this will include your full peer review and any attached files.

Reviewer #1: **Yes: **Guillermo B. Willis

Reviewer #2: No

---

## [Author Response · Author response to Decision Letter 0]

3 Nov 2020

October 3rd, 2020

RE: PONE-D-20-23794, “The bright side of pessimism: Promoting wealth redistribution under (felt) economic hardship”

Dear Dr. Miller,

Thank you very much for your constructive feedback on the manuscript “The bright side of pessimism: Promoting wealth redistribution under (felt) economic hardship” (PONE-D-20-23794) co-authored with Anne Maass and Annalisa Robbiani. We appreciate the opportunity to address the concerns by the reviewers, and for all the work that you and the reviewers put into this manuscript. We took the comments from the reviewers and your editorial feedback very seriously and addressed all the points in your Decision letter and the enclosed reviews. Below we will detail the changes that we made in the order in which they appear in the Decision letter.

EDITOR

COMMENT: Please ensure that your manuscript meets PLOS ONE's style requirements, including those for file naming.

REPLY: As requested, we have checked PLOS ONE’s style requirements, including those for file naming, and we believe that the revised version of the manuscript meets all the requirements.

COMMENT: Please change “female” or “male” to “woman” or “man” as appropriate, when used as a noun

REPLY: As suggested, we have changed “female” and “male” to “woman” and “man” when used as nouns (see pages 8 and 14 of the present revised version of the manuscript)

REVIEWER 1

COMMENT: OVERALL RATIONALE OF THE MANUSCRIPT 

I think this paper does an important contribution when examining the effects on the redistribution measure. Thus, I think it will be better to focus the rationale of the manuscript—a bit more—around this effect. The differences between personal optimism and collective pessimism, and how this relates with sense of control, are interesting research questions, but they have already been examined in the literature. Hence, I think it will be better to focus more the introduction around the redistribution research question and maybe put a bit aside the other questions—although it will still be important to include them in the manuscript, of course.

REPLY: We have added a paragraph in the first page of the manuscript (page 2), in which we discuss that, according to many experts, progressive taxation plays a key role in redistribution. However, lay people (and especially the poor), although typically in favor of redistribution, generally fail to support this strategy: “Among the different strategies proposed by economists and organizations alike to reduce the economic gap, progressive taxation occupies a central place. For instance, the 2020 annual report of Oxfam, entitled “Public Good or Private Wealth ”, proposes to “end the under-taxation of rich individuals and corporations”, to “tax wealth and capital at fairer levels”, to “stop the race to the bottom on personal income and corporate taxes”, and to “eliminate tax avoidance and evasion by corporations and the super-rich” [14]. Likewise, the World Economic Forum recently argued that “the introduction of a tax on passive income and wealth is essential in a world where individuals are wealthier than nations” [15]. And the International Monetary Fund has recently argued that the Covid-19 pandemic has made it ever more important “to move towards a fairer and more equitable taxation of economic activities at the global level” [16]. Although progressive taxation is considered a pillar of redistribution by many experts, lay people, though desiring a fairer society at an abstract level, rarely support increasing progressivity of income, corporate, or inheritance taxes. Thus, there is a remarkable contrast between the abstract quest for reducing inequality and the failure to support concrete redistribution strategies such as progressive taxation [17], which is the focus of the present research. Ironically, those social classes that would benefit most from such concrete redistribution strategies are often particularly reluctant to support them [18].”

Moreover, to give the taxation task a more prominent position in the paper, we also divided the former Hypothesis 5 into two hypotheses, one (Hypothesis 4) regarding the effects of mind-set and manipulated socio-economic status on support for progressive taxation, the other (Hypothesis 5) regarding the moderated mediation model (see page 13).

COMMENT: Part of this is also related with how the mediations are interpreted. I think that mediations provide preliminary information about the examined processes, but it will be important to corroborate these findings using experimental methodology (see Fiedler, Harris, & Schott, 2018, JESP; Spencer, Zanna, & Fong, 2005, JPSP). Thus, I think the authors may present this limitation of the mediation analyses in the general discussion. At the same time, they could give greater importance to the main (and interaction) effects of their experimental manipulations

REPLY: We thank Reviewer 1 for raising these important suggestions. Taking into account common misunderstanding of mediation analyses, we have revised the Discussion section of Study 1 also using the past tense, which restricts findings to the current study and not to a generalizable law. Moreover, we discuss this limitation also in the General Discussion (pages 49 and 50: “First of all, our results can provide only preliminary information about the mediating role of personal control. On the one side, it remains unclear why control beliefs mediated the relation between mind-set condition and expected economic, but not general, risks despite the fact that participants perceived greater control over personal than over collective risks in both cases. On the other side, even in the case of economic risks, we can not exclude the possibility that additional mediators may have played a role. Drawing on literature, perceived control seemed to be the most plausible psychological mechanism that could explain the asymmetry between personal optimism and collective pessimism [64, 86-89]. However, many mechanisms contribute to producing personal optimism [84] and, therefore, an exhaustive analysis aimed at identifying the true mediator would have to rely on diagnostic tests of mediation against other candidates that we did not assess in the present work. For example, individuals are more optimistic about their personal than their collective future also because they feel it is under their own personal responsibility [89]. It is possible that this and possibly additional psychological processes may have contributed to the relation between mind-set and expected economic and general risks above and beyond personal control. Therefore, we deem it necessary for future research to extend the present findings investigating other psychological mechanisms in addition to personal control.” 

Moreover, we have revised the whole manuscript, also trying to highlight the main effects and, in particular, the interaction effects of our experimental manipulations (including the introduction of a separate hypothesis). 

COMMENT: WHERE IS THE MAIN DIFFERENCE?

All in all, the authors argue that participants who were in the low class/collective pessimism condition where those who supported more the progressive redistribution policy. Although this seems true in Study 2, I am not so sure this also happens in Study 1. Specifically, in Study 1 the means show that participants in this group (M = 29.24) were very similar to those participants assigned to the high class/collective pessimism (M = 27.06, and to the high class/personal optimism (M = 28.00) condition. Thus, in Study 1, participants that tended to differ from the other three conditions were those assigned to the low class/personal optimism condition (M = 23.88)—not those assigned to the low class/collective pessimism condition. I think this is important because, although the interaction effect is replicated in Study 2, the direction of the means may be telling a different story. The authors may check if these are really different patterns using planned contrasts.

If there are different patterns, one way of knowing which of the patterns is more consistent—if participants in the collective/low class or participants in the personal/low class are the ones driving the effect— may be to run a pooled data (or integrative data) analysis. Performing the analysis with the pooled sample may answer the question about which group is different from the others. 

REPLY: Following the reviewer’s suggestion we ran a pooled data analysis on scores of progressivity of taxation. Results revealed reduced support for tax-based redistribution in the personal mind-set and increased support for tax-based redistribution in the collective mind-set condition. These findings are reported on pages 44 and 45: “Although both studies confirmed the predicted interaction between manipulated socio-economic status and mind-set on support for tax-based redistribution, the result pattern was slightly different. In both studies, those who felt relatively well-off were unaffected by the mind-set manipulation, but the effect of mind-set on those who were made to feel disadvantaged differed across studies. These participants reduced their support for redistribution when in a personal mind-set in Study 1 but increased their support for redistribution when in a collective mind-set in Study 2. To understand whether personal mind-set increases or whether collective mind-set reduces support for redistribution among the poor, or whether both are operating to a similar extent, we conducted a pooled data analysis. The 2 (manipulated socio-economic status) x 2 (mind-set) x 2 (Study) ANOVA confirmed the interaction between mind-set and manipulated socio-economic status, F(1, 682) = 13.10, p < .001, �2p = .02, which, importantly, was not moderated by Study 1 vs. 2, F(1, 682) = .50, p = .48, �2p = .001. Compared to participants who felt relatively rich, those who felt relatively poor proposed a less progressive taxation when in a personal mind-set, t(343) = 2.66, p = .008, but a more progressive taxation when in a collective mind-set, t(343) = -2.33, p = .02. Thus, depending on mind-set condition, redistribution attitudes of those who felt relatively poor were greatly polarized.”

We also discuss these results in the Discussion section on page 50: “….a comparison with those who felt relatively rich (see pooled data analysis) suggests that both processes may be operating, given that those who felt poor showed less support for progressive taxation than the “rich” counterparts in the collective, but more support in the personal mindset condition. Thus, support for progressive taxation of participants who were made to feel relatively poor were greatly polarized. On the one side, collective pessimism motivated those who felt poor to endorse concrete redistribution strategies. On the other side, although not originally hypothesized, personal optimism hampered their wish for tax-based redistribution. In line with prior theorizing [18], it is plausible to argue that those who felt relatively poor and were considering their personal (rather than collective) risks, either made stronger internal attributions or justified the system more, resulting in less support for redistribution. It is up to future research to investigate the exact underlying reasons of this polarized view on progressive taxation among the poor.”

COMMENT: WHAT IS THE COLLECTIVE PESSIMISM/PERSONAL OPTIMISM MANIPULATION CHANGING?

One of my biggest doubts was about the collective pessimism vs. personal optimism manipulation. I am not sure about what this manipulation is exactly comparing. In these studies, is it collective pessimism that make participants more aware of inequalities and that is why they tend to support more redistribution? Or is it that thinking about their (seemingly positive) personal outcomes make them less prone to redistribute money? Said otherwise, I think that using a control group will be important to know where is the effect. In fact, the authors say that participants want to redistribute more when they are deprived and when they have “Awareness that things will get worse”. Thus, if they believe that this is what is driving the effect, I think it may be a good idea to directly manipulate this process, instead of doing the collective pessimism vs. personal optimism contrast. Authors may compare a situation in which “things will go collectively worse” against a situation in which they will stay the same. Alternatively, the authors may argue that they will not find differences when comparing “when things will go collectively worse” with “when things will stay collectively the same”, because what is important is to believe that you will be better off (or not) in the future. In this case, I think it will be important to orthogonally manipulate both variables—and perform a 2 (High and low in collective pessimism) x 2 (High and low in personal optimism) experiment.

I think that a paper with any of these two studies may be more informative. However, if the authors are not able to run the study again, it may be important to include this in the limitations of the study—that, given that the authors do not have a control group, you could not know if the effect is driven by collective pessimism or personal optimism.

REPLY: The Reviewer raises two interrelated critiques concerning the mind-set manipulation, both of which we addressed in the discussion section on pages 50 and 51. On the one side, the reviewer argues that only a no-mind-set control condition would allow to quantify the relative weight of the two processes and to disentangle increased support due to collective optimism from decreased support due to personal optimism among those who felt relatively poor. We have now acknowledged this limitation and at the same time argued that our pooled data analysis, comparing those who were made feel rich vs. poor, suggests that both processes seem to be operating. “However, one could argue that, in the absence of a no-mind-set control condition, our experimental design does not allow to disentangle with certainty whether it is collective pessimism that makes relatively poor participants more prone to support redistribution policies, or whether personal optimism inhibits their support for redistribution. Although we acknowledge this limit, a comparison with those who felt relatively rich (see pooled data analysis) suggests that both processes may be operating, given that those who felt poor showed less support for progressive taxation in the collective, but more support in the personal mind-set condition. Thus, support for progressive taxation of participants who were made to feel relatively poor were greatly polarized. On the one side, collective pessimism motivated those who felt poor to endorse concrete redistribution strategies. On the other side, although not originally hypothesized, personal optimism hampered the wish for tax-based redistribution among those who felt relatively poor. In line with prior theorizing [18], it is plausible to argue that those who felt relatively poor and were considering their personal (rather than collective) risks, either made stronger internal attributions or justified the system more, resulting in less support for redistribution. It is up to future research to investigate the exact underlying reasons of this polarized view on progressive taxation among the poor.” (page 50).

On the other side, the reviewer proposed a direct manipulation of optimism and pessimism, which was addressed in the subsequent paragraph (pages 50 and 51): “Another extension of our collective vs. personal mind-set manipulation may be to manipulate the optimistic or pessimistic outlook directly, for instance by informing participants that things will (or will not) get worse collectively (or personally). However, such information-based manipulations are often insufficient to induce changes in tax preferences [25]. It therefore remains to be seen whether information-based interventions work as well or better than the indirect approach chosen here, in which people generate their own (optimistic or pessimistic) outlook. Our work, as well as literature, suggests that it is the focus on the collective per se that makes people pessimistic about the future, and it is the focus on the self per se that makes people unrealistically optimistic about the future, thus increasing and reducing, respectively, support for redistribution among the relatively poor. A systematic comparison of the two approaches may be useful both from a theoretical and applied point of view.”

COMMENT: COVID-19

The authors ran Study 2 during the hardest times of the COVID-19 pandemic. As a researcher doing studies about the social psychology of economic inequality, I found that most of the studies I ran during these months tend to present unusual results. Although it is clear that it did not influence much the effects of this study, as the authors were able to replicate their finding, I was wondering whether the authors controlled or asked some questions related with the pandemic. Or if it influenced in some way participants’ responses; for instance, the collective pessimism manipulation may be stronger after the pandemic than before.

REPLY: We find this argument very interesting and we have addressed it on page 52: “Finally, the fact that our second study was run during one of the peaks of the Covid-19 pandemic may have affected our results in unknown ways. Many researchers have observed “anomalous” responses to surveys administered during the pandemic, so this possibility can not be excluded with certainty. Specifically, participants may have experienced worse mood and less control over their lives during the pandemic. Also, the very issue of economic inequality may have become less central due to the overwhelming health concerns that took center stage in the media (and in people’s minds) at that point in time. Comparing pre-experimental mood across studies, we observed a slightly better mood among participants of (pre-Covid) Study 1 (M = 6.53, SD = 1.84) than among those of Study 2 (M = 6.21, SD = 1.78), although differences were small considering the 1-10 scale on which mood was assessed. However, participants in Study 2 reported a somewhat greater sense of personal control (M = 4.91, SD = 1.33) than those in (pre-Covid) Study 1 (M = 4.54, SD = 1.42), Thus, there is no consistent pattern suggesting that participants’ responses in Study 2 may have been negatively impacted by the massive disruption experienced during Covid-19. More importantly, the similarity of result patterns obtained in Studies 1 (pre-Covid-19) and 2 (during Covid-19) for all hypotheses, including the complex moderated mediation model, argues against a reliable interference of Covid-19 in this set of studies.”

Finally, Reviewer 1 specifically argues that “the collective pessimism manipulation may be stronger after the pandemic than before”. As far as the low status participants are concerned, our data are in line with this interpretation. However, the fact that the mind-set manipulation differed across studies makes it difficult to draw clear conclusions in this regard. We therefore decided not to make this argument in the text, but we are prepared to add it if you consider it useful. 

COMMENT: Hypothesis 4 may be rewritten. It was not very clear for me that this was a “moderated mediation” hypothesis. This, of course, may not be a deviation from the preregistration as the hypothesis will be the same—and only the wording changes.

REPLY: The hypothesis, now Hypothesis 5, was reformulated (see page 13) as follows: “Hypothesis 5: collective mind-set would reduce participants’ perceived control, which, in turn, should increase pessimism about collective economic risks. Greater pessimism about collective economic risks would lead to greater support for progressive taxation only for participants who felt relatively poor.” To facilitate comprehension we also added a Figure (Fig 1) representing the predicted moderated mediation model.

COMMENT: I tried to look for the preregistration, but I was not able to do so. Please include a link to the preregistration in the paper

REPLY: We apologize for this omission. The link is now reported on page 32.

REVIEWER 2

COMMENT: I think the idea is worthy to study, however the analyses and results are not presented in an easy way. That makes it difficult to interpret the main results and assess whether there is something missing in the analysis that may be important. Also, the comparison between analyses it is not straightforward. I would recommend using tables to present the experimental design, the descriptive statistics and the main analyses. Showing the average, standard deviation and p-values in the main text makes it difficult to follow the main argument of the paper

REPLY: We thank Reviewer 2 for this remark. This criticism was addressed in two ways:

1. We moved the description of the study design at the beginning of the method section (pages 13 and 14) to clarify that this was a 2x2 factorial design: “The experiment was created using the software Survey Monkey and designed in such a way as to avoid any missing data. The experiment consisted of 2 X 2 factorial design. The independent variables of interest were the manipulated socio-economic status (low or high) and the mind-set (personal or collective), which were manipulated between-participants to obtain 4 levels (low socio-economic status and personal mind-set, low socio-economic status and collective mind-set, high socio-economic status and personal mind-set, high socio-economic status and collective mind-set) resulting in 4 surveys. The main dependent variables consisted of the two brief scales aimed at assessing participants’ personal control and expected risks, and the taxation task.”

2. Means and SD of all main variables of Study 1 and Study 2 were already reported in two tables in the prior version of the manuscript (Table 2 and Table 4 of the present revised version of the manuscript). These two Tables were maintained and expanded to include, for each variable, also means and SD as a function of mind-set condition. Moreover, for both studies, zero-order correlations among main study variables (Table 1 and Table 3 of the present revised version of the manuscript) now include also participants’ family characteristics (family size, number of family members below 18 years of age, number of people contributing to the family income). These changes allowed us to include averages, standard deviations and p-values in the main text only if strictly necessary. Finally, to facilitate the comprehension of the main argument of the paper, for both studies the headings for the sub-sections of the Main Analyses section include also the hypothesis being tested (e.g., Hypothesis 1: Personal control), which is also reported at the beginning of the relevant paragraphs. Finally, as mentioned above, we added a figure representing the moderated mediation model on page 13, which should facilitate comprehension of this complex prediction. 

COMMENT: Also, I would recommend to add sociodemographic variables as control in the different analyses presented in the text.

REPLY: We have addressed this concern on pages 24 and 25 and pages 38-39 of the present revised version of the manuscript. Specifically, we have added to the preliminary analyses of the Study 1 (pages 24-25) and Study 2 (pages 38-39), a series of multiple regression analyses on our main dependent variables (i.e., personal control, economic risks, general risks, and progressivity of taxation), using manipulated socio-economic status (low = 0, high = 1), mind-set (personal = 0, collective = 1), gender, age, education, occupation, and their interactions (second block) as predictors. In Study 1, “the model predicting participants’ scores of personal control revealed a significant Education x Mind-set interaction, b = .25, t(290) = 2.72, p = .007: in the collective, but not personal, mind-set participants with the lowest formal qualification perceived less personal control (M = 2.87, SD = 1.55) than those with a high school diploma (M = 4.13, SD = 1.41, t(93) = -3.49, p = .001) and those with a university degree (M = 4.47, SD = 1.43; t(182) = -4.29, p < .001), whereas no differences in perceived personal control emerged between participants with a high school diploma and those with a university degree (t(137) = -1.43, p > .15). However, participants with the lowest formal qualification, as well as those with high school diploma and with a university degree reported greater personal control in the personal than in the collective mind-set condition (lowest formal qualification: t(33) = 3.26, p = .003; high school diploma: t(135) = 4.76, p < .001; university degree: t(132) = 2.07, p = .04), thus suggesting that the manipulation was effective for all subjects. No other interaction effect emerged (ps > .08)”. In Study 2, “in the model predicting participants’ scores of personal control, the effect of Gender x Mind-set was significant, b = .25, t(371) = 2.93, p = .004, indicating that men perceived greater personal control than women in the personal mind-set (men: M = 5.6, SD = 0.93; women: M = 5.3, SD = 0.87; t(196) = 2.78, p = .006), whereas women felt more personal control than men in the collective mind-set (women: M = 4.6, SD = 1.48; men: M = 4.2, SD = 1.49; t(184) = -1.95, p = .05). However, both men and women reported greater personal control in the personal than in the collective mind-set condition (men: t(167) = 7.60, p < .001; women: t(213) = 4.18, p < .001), thus suggesting that the manipulation was effective for both genders. Importantly, no other interaction effect was found (ps > .07)” (pages 34-35). Given these results sociodemographic variables were not discussed further in the main analyses of both studies.

COMMENT: Why do the authors restrict the sample to participants with a personal monthly income between 1200 and 1800 euros? The authors should justify this decision and explain how it could affect the results. It is difficult to think how credible the socio-economic manipulation is as all participants are equally rich or poor in their real live. The high number of participants, in study 1 and also in study 2, that do not agree with the assigned position in the economic ladder would indicate that the manipulation is not working properly and it could have some important consequences in the results. Please, discuss these two points and explain their main consequences and how they may limit the scope of the results

REPLY: We have addressed the Reviewer 2’s concern on page 14 (as well as in the Discussion on page 48) by explaining in detail the rationale behind this choice: “We decided to limit the personal monthly income of our sample to this range because, according to a recent national study [97], Italians’ average monthly income is around 1500 euros. Importantly, we believed that the more homogeneous our sample was with respect to monthly income range, the more the manipulation of participants’ socio-economic status would be likely to be effective, thus allowing us to avoid the potentially confounding variables that typically covary with individuals’ actual socio-economic status (e.g., political orientation, tax attitudes, system justification).” 

Also, please note that the number of participants that did not agree with the assigned position in the socio-economic ladder does not seem excessively high, given that only 12% of participants in Study 1 and 11% of participants in Study 2 were excluded. To make this clear we now report percentages (in addition to frequencies) in the manuscript on pages 21 and 33, respectively.

In sum, we believe that if we had participants with different personal monthly incomes the socio-economic manipulation would be significantly less credible and we would not have the opportunity to avoid a number of potentially confounding variables.

In summary, we are very grateful to you and the reviewers for the effort that went into the review of our manuscript. We found the editorial feedback to be very helpful in revising the paper, and we believe that the suggested changes have significantly increased the quality of our paper. We believe that now the manuscript is much stronger and hopefully ready for publication in PLOS ONE, and we look forward to hearing your reactions. If you have any questions, please do not hesitate to contact us. 

Best wishes

Silvia Galdi, Anne Maass, and Annalisa Robbiani

---

## [Decision Letter · Decision Letter 1]

23 Nov 2020

The bright side of pessimism: Promoting wealth redistribution under (felt) economic hardship

PONE-D-20-23794R1

Dear Dr. Galdi,

We are pleased to inform you that your manuscript has been judged scientifically suitable for publication and will be formally accepted for publication once it meets all outstanding technical requirements. When you submit the final version of the paper, please add the new footnote suggested by reviewer 1.

Within one week, you will receive an e-mail detailing the required amendments. When these have been addressed, you will receive a formal acceptance letter and your manuscript will be scheduled for publication.

Kind regards,

Luis M. Miller, Ph.D.

Academic Editor

PLOS ONE

Reviewers' comments:

Reviewer's Responses to Questions

**Comments to the Author**

1. If the authors have adequately addressed your comments raised in a previous round of review and you feel that this manuscript is now acceptable for publication, you may indicate that here to bypass the “Comments to the Author” section, enter your conflict of interest statement in the “Confidential to Editor” section, and submit your "Accept" recommendation.

Reviewer #1: All comments have been addressed

Reviewer #2: All comments have been addressed

2. Is the manuscript technically sound, and do the data support the conclusions?

Reviewer #1: Yes

Reviewer #2: Yes

3. Has the statistical analysis been performed appropriately and rigorously? 

Reviewer #1: Yes

Reviewer #2: Yes

4. Have the authors made all data underlying the findings in their manuscript fully available?

Reviewer #1: Yes

Reviewer #2: Yes

5. Is the manuscript presented in an intelligible fashion and written in standard English?

Reviewer #1: Yes

Reviewer #2: Yes

6. Review Comments to the Author

Reviewer #1: I thank the authors for addressing all my comments.

My last recommendation will be to add (maybe in a footnote?) a small comment after the preregistration, saying that the number of the hypothesis in the text do not correspond with the number of the preregistered ones. I agree with rewording and renumbering the hypotheses, but I think it will be important to add such a clarification.

Guillermo B. Willis

University of Granada

Reviewer #2: (No Response)

7. PLOS authors have the option to publish the peer review history of their article (what does this mean?). If published, this will include your full peer review and any attached files.

Reviewer #1: **Yes: **Guillermo B. Willis

Reviewer #2: No

---

## [Editor Report · Acceptance letter]

1 Dec 2020

PONE-D-20-23794R1 

The bright side of pessimism:
Promoting wealth redistribution under (felt) economic hardship 

Dear Dr. Galdi:

I'm pleased to inform you that your manuscript has been deemed suitable for publication in PLOS ONE. Congratulations! Your manuscript is now with our production department. 

Kind regards, 

on behalf of

Dr. Luis M. Miller 

Academic Editor

PLOS ONE